# Differential gene expression in peripheral leukocytes of pre-weaned Holstein heifer calves with respiratory disease

Lily A. Elder☉, Holly R. Hinnant☉, Chris M. Mandella‡, Rachel A. Claus-Walker‡, Lindsay M. Parrish‡, Giovana S. Slanzon¤‡, Craig S. McConnel ✆*☉

Department of Veterinary Clinical Sciences, Field Disease Investigation Unit, Washington State University, Pullman, Washington, United States of America

☉ These authors contributed equally to this work.
¤ Current address: Department of Tropical Plant and Soil Sciences, University of Hawai'i at Mānoa, Honolulu, Hawaii, United States of America
‡ CMM, RAC, LMP and GSS also contributed equally to this work.
* cmcconnel@wsu.edu

**Data Availability Statement:** All relevant data are within the paper and its Supporting information files.

## Abstract

Bovine respiratory disease (BRD) is a leading cause of calf morbidity and mortality, and prevalence remains high despite current management practices. Differential gene expression (DGE) provides detailed insight into individual immune responses and can illuminate enriched pathways and biomarkers that contribute to disease susceptibility and outcomes. The aims of this study were to investigate differences in peripheral leukocyte gene expression in Holstein preweaned heifer calves 1) with and without BRD, and 2) across weeks of age. Calves were enrolled for this short-term longitudinal study on two commercial dairies in Washington State. Calves were assessed every two weeks throughout the pre-weaning period using clinical respiratory scoring (CRS) and thoracic ultrasonography (TUS), and blood samples were collected. Calves were selected that were either healthy (n = 10) or had BRD diagnosed by CRS (n = 7), TUS (n = 6), or both (n = 6) in weeks 5 or 7 of life. Three consecutive time point samples were analyzed for each BRD calf consisting of PRE, ONSET, and POST samples. Nineteen genes of interest were selected based on previous gene expression studies in cattle: *ALOX15*, *BPI*, *CATHL6*, *CXCL8*, *DHX58*, *GZMB*, *HPGD*, *IFNG*, *IL17D*, *IL1R2*, *ISG15*, *LCN2*, *LIF*, *MX1*, *OAS2*, *PGLYRP1*, *S100A8*, *SELP*, and *TNF*. Comparisons were made between age and disease time point matched BRD and healthy calves as well as between calf weeks of age. No DGE was observed between diseased and healthy calves; however, DGE was observed between calf weeks of age regardless of disease state. Developmental differences in leukocyte gene expression, phenotype, and functionality make pre-weaned calves immunologically distinct from mature cattle, and early life shifts in calf leukocyte populations likely contribute to the age-related gene expression differences we observed. Age overshadows disease impacts to influence gene expression in young calves, and immune development progresses upon a common trajectory regardless of disease during the preweaning period.

**Funding:** This project was supported by Agriculture and Food Research Initiative Competitive Grant no. 2021-68014-34144 from the USDA National Institute of Food and Agriculture, and the USDA National Institute of Food and Agriculture, Animal Health & Disease Research Capacity Grant project 1014680. This work was also supported by funds from the USDA National Institute of Food and Agriculture, Animal Health and Disease Research Program project NI21AHDRXXXXG028. There was no additional external funding received for this study. The funders had no role in study design, data collection and analysis, decision to publish, or preparation of the manuscript.

**Competing interests:** The authors have declared that no competing interests exist.

## Introduction

Bovine respiratory disease (BRD) is a leading cause of calf morbidity and mortality in both beef and dairy sectors of the United States cattle industry resulting in financial losses and compromised welfare [1–3]. BRD is a multifactorial disease complex with varied etiology including viral and bacterial pathogens as well as environmental and husbandry factors, and it ultimately manifests as bronchopneumonia [4, 5]. BRD is currently managed with prophylactic vaccination and antimicrobial treatment of clinically affected animals; however, morbidity and mortality attributed to this disease complex have increased over the past decades [6–10].

The immune response of individual animals plays a significant role in determining susceptibility to and severity of BRD as the host inflammatory process is largely responsible for the clinical signs and tissue damage that leads to negative sequelae [11–13]. Differential gene expression (DGE) provides detailed insight into individual immune responses and can illuminate enriched pathways and biomarkers that contribute to disease susceptibility and outcomes [10, 14, 15]. Pathways associated with innate immunity were found to be consistently enriched in bronchial lymph node tissue from single BRD pathogen challenged cattle suggesting the potential for a diagnostic gene expression signature of BRD [16, 17]. Differential expression of genes related to innate immune function were also demonstrated in peripheral leukocytes from both beef and dairy breeds with experimentally induced or naturally occurring BRD [18–20]. Additionally, differential expression of genes involved in inflammatory regulation has been found prior to clinical disease in cattle that were subsequently diagnosed with BRD, and expression differences were associated with the eventual severity of disease [14, 15, 21]. These findings further indicate the possibility of identifying biomarkers for use in early disease diagnosis, susceptibility screening tools, or novel targeted therapeutic modalities.

Prior gene expression studies related to naturally occurring BRD have been conducted solely in post-weaned beef cattle. Baseline transcriptomic differences have been demonstrated between cattle breeds highlighting the necessity of testing these findings in dairy cattle breeds [22, 23]. Additionally, considerable differences in gene expression based on age have been demonstrated in human studies [24, 25]. This indicates the need to test the findings from feedlot and adult cattle studies in pre-weaned Holstein calves.

In this study we analyzed the expression of 19 genes in peripheral leukocytes of preweaned Holstein calves with and without BRD. Selected genes were previously shown to be differentially expressed in cattle with BRD and other common diseases of dairy cattle [14, 16–19, 21, 26–30]. Genes were chosen based on the antibacterial or antiviral action of their protein products or their role in the inflammatory process with the goal of identifying indicators of early or subclinical disease, markers of innate disease resistance and tolerance, and common expression patterns across disease states.

The aims of this study were to investigate differences in peripheral leukocyte gene expression in Holstein preweaned calves 1) with and without BRD, and 2) across weeks of age. We hypothesized that both disease status and temporal changes would lead to differential expression of the selected genes.

## Materials and methods

### Ethics statement

The research protocol was reviewed and approved by the Institutional Animal Care and Use Committee of Washington State University (IACUC protocol #6859).

## Cohort enrollment and management

For this short-term longitudinal study, a cohort of 60 Holstein heifer calves was enrolled on two commercial dairies (Farms A and B) in Washington State with a convenience sample of 30 individuals per farm selected based on age and adjacent housing. Calves were enrolled at one week (±4 days) of age and sampled every other week through 11 weeks of age spanning May-August 2021.

Both dairies housed calves in individual pens separated by removable, solid panels within an open-sided, covered housing unit containing a total of 60 calves (30 per side). Calves were grouped in pairs at nine weeks of age on Farm A by removing solid panels between each pair of calves, whereas calves remained individually housed throughout the study on Farm B. On both farms calves received 1 gallon of colostrum (Brix refractometer ≥ 22%) via oral intubation within 30 minutes of birth and a second gallon (Brix refractometer ≤ 22%) 8–12 hours later. All the calves were bucket fed and weaned at eight weeks of age. On Farm A all calves received a single, intranasal dose of Inforce 3® (Zoetis Animal Health, Parsippany, NJ) at four weeks of age. On Farm B all calves received a single, intranasal dose of Nasalgen® 3 (Merck Animal Health, Kenilworth, NJ) at two weeks of age.

## Assessments and sampling

Clinical assessments (CRS) and thoracic ultrasonography (TUS) for BRD diagnosis were performed at each sampling with TUS assessment starting at the second sampling in week three of life. Calves were scored for clinical signs of respiratory disease using the Wisconsin Calf Health Scoring Chart [31]. The respiratory parameters in this system consist of nose, eyes, cough, and temperature, and they are scored using a 0–3 scale with 0 indicating normal and 1–3 indicating mild, moderate, or severe clinical signs. The navel, joint, ears and fecal parameters included in the scoring system also were assessed.

TUS was performed and lung lesions scored using the technique previously described by Ollivett and Buczinski [32]. Briefly, the lung fields were imaged by scanning the tenth through first intercostal spaces from dorsal to ventral on both sides of the calf. The seven peripheral lung lobes were scored using a 0–5 scale with 0 indicating healthy lungs, 1 indicating diffuse comet tailing, 2 indicating lobular consolidation of ≥1 cm but not full thickness, 3 indicating full thickness lobar consolidation of one lung lobe, 4 indicating full thickness lobar consolidation of two lobes, and 5 indicating full thickness lobar consolidation of three or more lobes. All clinical assessments and TUS examinations were performed under the supervision of the PI, McConnel, including secondary reviews of recorded video for all identified lung lesions. TUS was performed using IBEX® EVO® II and IBEX® PRO/r ultrasounds both with a L7HD linear probe for large animal exams (E.I. Medical Imaging, Loveland, CO).

Blood samples were collected at each sampling. Blood was collected from the jugular vein into Tempus Blood RNA Tubes (Thermo Fisher, Waltham, MA) and shaken vigorously before being placed on ice. Samples were refrigerated for up to 48 hours, frozen at -20˚ C for up to four weeks, and stored at -80˚ C.

## Sample selection for differential gene expression (DGE)

Calves were retrospectively selected for DGE analysis based on CRS and TUS scoring and farm treatment records. Individuals were categorized at each sampling as CRS- or CRS+ and TUS- or TUS+. CRS- was defined as a score of ≤ 1 in all the respiratory parameters (nasal, eyes, and cough). CRS+ was defined as a score of ≥ 2 in one or more of the respiratory parameters. Rectal temperature was not considered in the CRS definition due to its lack of specificity regarding respiratory disease, and because a period of abnormally hot weather occurred that

affected both farms equally during the sampling period (late June to mid-July) resulting in hyperthermia in otherwise healthy calves. Calves with a navel score of $\geq 2$ were not considered for DGE.

Considerable variation of lobular lung lesion sizes, number, and distribution was encompassed by a TUS score of 2; therefore, TUS- was defined as scores of $\leq 1$ and TUS+ was defined as lobar scores of $\geq 3$ (i.e., full thickness lobar consolidation of $\geq 1$ lung lobe) to eliminate the ambiguity of lesion severity associated with a lobular score of 2. Calves were then further categorized into one of the following four categories: CRS-/TUS- (healthy), CRS+/TUS-, CRS-/TUS+, and CRS+/TUS+. Calves were selected for DGE analysis that were CRS-/TUS- at one sampling and then progressed to one of the other categories at the next sampling indicating the onset of respiratory disease. A healthy comparison group was selected that remained CRS-/TUS- throughout the sampling period. Calves that were treated with antimicrobial and/ or anti-inflammatory drugs by the farm personnel prior to the ONSET sample were not considered for DGE analysis.

Three consecutive samples were analyzed for each calf to assess the full progression of disease. These samples spanned either weeks 3–7 or 5–9 of life depending on the timing of the onset of respiratory disease. The three time points were defined as follows. At the first sample time point (PRE) the calf was CRS- and TUS-. At the second time point (ONSET) the calf fit one of the three BRD categories (CRS+/TUS-, CRS-/TUS+, or CRS+/TUS+). The third time point (POST) was not defined by diagnostic parameters but was intended to capture the residual effects of the disease event and the continuation or resolution phase of the inflammatory cascade. PRE was either week 3 or 5 of life, ONSET was either week 5 or 7, and POST was either week 7 or 9. This timeframe was chosen in order to span the pre-weaning period. Four consecutive samples spanning weeks 3–9 of life were analyzed for most of the healthy calves to maximize the numbers of healthy comparisons at each age time point. The distribution of individuals in each disease category by week of age at ONSET are shown in Fig 1. The individual CRS and TUS scores for each calf by sampling week can be found in the S1 Table.

The distribution of individuals in each BRD category by week of age at ONSET is shown. Calves were selected that had the onset of BRD in either week 5 or 7 of life. PRE and POST samples were also analyzed for each individual with PRE occurring two weeks before ONSET (week 3 or 5) and POST occurring two weeks after ONSET (week 7 or 9).

## Sample size

Previous DGE studies have shown that a minimum of eight animals should be targeted per group (diseased, healthy), with the sample size based on expectations for a multivariate analysis estimating the log fold-change in a target gene's DGE following a negative binomial distribution [33]. Historical precedent on the participating dairies suggested that on average 15% of calves would be diagnosed and treated for respiratory disease through 12 weeks of age. Therefore, expectations were for quality RNA samples from a minimum of 9 calves with evidence of respiratory disease (CRS+ and/or TUS+), and at least equal numbers of consistently healthy calves.

## Gene selection

Nineteen genes of interest were selected based on previous studies of gene expression in cattle: *ALOX15*, *BPI*, *CATHL6*, *CXCL8*, *DHX58*, *GZMB*, *HPGD*, *IFNG*, *IL17D*, *IL1R2*, *ISG15*, *LCN2*, *LIF*, *MX1*, *OAS2*, *PGLYRP1*, *S100A8*, *SELP*, and *TNF* [14, 16–19, 21] (Table 1). All the genes were found to be differentially expressed in two or more studies focused on BRD, and at least one study each that used RNA extracted from whole blood samples. Ten of the genes

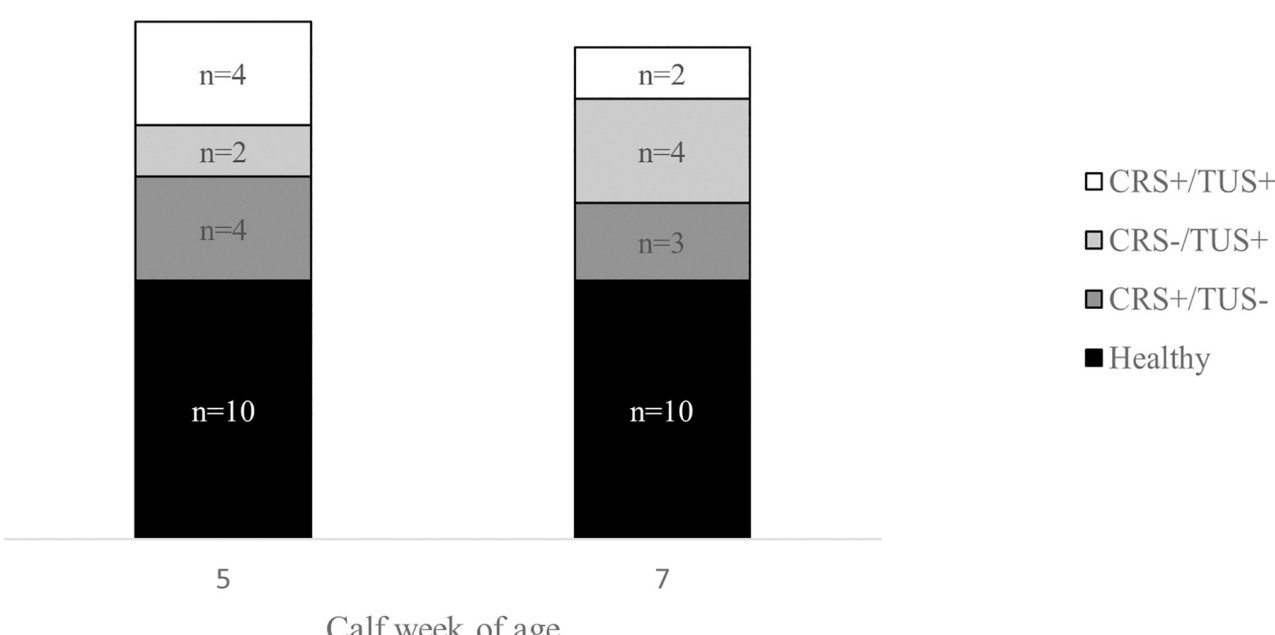

**Fig 1. Age and number of calves at the onset of disease for each of the BRD categories and for age-matched healthy calves.**

(*CATHL6*, *CXCL8*, *DHX58*, *IFNG*, *IL17D*, *LCN2*, *LIF*, *PGLYRP1*, *SELP*, and *TNF*) also were differentially expressed in dairy cattle diagnosed with metritis, mastitis, or Johne's Disease [26–30], and these genes have the potential to provide insight into conserved elements bridging disease states and target tissues in diverse dairy populations. Genes were prioritized that have strong evidence for their role in amplification or attenuation of inflammatory processes, the antibacterial or antiviral actions of their protein products, or their involvement in disease resistance and tolerance mechanisms. Five internal control genes (*ACTB*, *GOLGA5*, *OSBPL2*, *SMUG1*, and *YWHAZ*) were also selected based on their previously demonstrated suitability as controls in bovine peripheral leukocytes [29].

## Leukocyte RNA extraction

Extraction of RNA from peripheral blood leukocytes was accomplished using the Tempus Spin RNA isolation kit (Thermo Fisher, Waltham, MA). RNA was quantified using a Nano-Drop One Spectrophotometer (Thermo Fisher Scientific, Waltham, MA USA). Samples were selected at random to test RNA integrity using a Qubit 4 Fluorometer (Invitrogen, Waltham, MA, USA). 100 μl of eluted RNA was stored at -80˚C until submission for DGE analysis. Prior to submission, RNA was diluted to a working concentration of 20 ng/μl (15.0–35.3 ng/μl) measured using a NanoDrop Spectrophotometer (Thermo Fisher Scientific, Waltham, MA). RNA samples were processed by the NanoString Technologies Proof of Principle Services in Seattle, WA. Gene expression analysis of the 19 target genes and five housekeeping genes was performed using the Nanostring System (Nanostring Technologies, Seattle, Washington, USA), as previously described [29]. Gene expression was measured using a custom CodeSet for the selected genes in the Nanostring nCounter Analysis System 2.0 (NanoString Technologies, Seattle, WA). The NanoString technology uses a digital color-coded barcode tag with single-

**Table 1. Genes of interest examined for differential expression in Holstein heifer calf peripheral leukocytes.**

| Gene symbol | Protein name | Biological function | Biological function and process |
|---|---|---|---|
| ALOX15 | Polyunsaturated fatty acid lipoxygenase | Lipid metabolism | Production of specialized pro-resolving mediators for inflammatory resolution |
| BPI | Bactericidal permeability increasing protein | Antibacterial | Antimicrobial against Gram negative organisms |
| CATHL6 | Cathelicidin 6 | Antibacterial | Antimicrobial against Gram positive, Gram negative, and fungal organisms |
| CXCL8 | Interleukin-8 | Inflammatory response | Chemotactic factor for neutrophil attraction and activation |
| DHX58 | ATP-dependent RNA helicase | Antiviral | Regulation of antiviral signaling |
| GZMB | Granzyme B | Apoptosis | Protease present in the granules of cytotoxic T-cells and NK-cells involved in induction of cytolysis and apoptosis |
| HPGD | 15-hydroxyprostaglandin dehydrogenase | Lipid metabolism | Breakdown of pro-inflammatory prostaglandins and cytokines |
| IFNG | Interferon gamma | Inflammatory response | Antiviral activity and immunoregulatory functions |
| IL17D | Interleukin 17D | Inflammatory response | Pro-inflammatory cytokine |
| IL1R2 | Interleukin 1 receptor type 2 | Inflammatory response | Binds and inactivates the pro-inflammatory cytokine IL1B |
| ISG15 | Ubiquitin like protein | Antiviral | Binds intracellular viral proteins inducing recognition and destruction by the innate immune system |
| LCN2 | Neutrophil gelatinase-associated lipocalin | Apoptosis | Binds and sequesters iron thereby inducing apoptosis and inhibiting bacterial proliferation |
| LIF | Leukemia inhibitory factor | Inflammatory response | Cytokine that influences leukocyte differentiation and acute phase protein synthesis |
| MX1 | Interferon-induced GTP-binding protein | Antiviral | Impedes transcription of the viral genome by inhibiting viral polymerase |
| OAS2 | 2'-5'-oligoadenylate synthase 2 | Antiviral | Detects double-stranded RNA and induces RNA degredation thereby inhibiting viral genome replication |
| PGLYRP1 | Peptidoglycan recognition protein 1 | Antibacterial | Antimicrobial against Gram positive organisms |
| S100A8 | Protein S100-A8 | Inflammatory response | A central regulator of the inflammatory response with activity including leukocyte recruitment and activation, zinc sequestration, and apoptosis |
| SELP | P-selectin | Cell adhesion | Mediates the interraction of circulating leukocytes with endothelial cells allowing leukocyte tethering and rolling |
| TNF | Tumor necrosis factor | Inflammatory response | Cytokine with a central role in the inflammatory response and apoptosis |

molecule imaging that can detect and count hundreds of unique transcripts per reaction. The nCounter run was performed at NanoString using a MAX analyzer, with 555 maximum possible fields of view. There were two consecutive normalizations: 1) A Positive Control Normalization took into account the linearity of the positive controls, using the geometric mean of the positive controls to compute the normalization factor. 2) CodeSet Content (HouseKeeping Normalization) used the geometric mean of the designated housekeeper genes to compute a normalization factor.

## Gene expression analysis

The data analysis was performed using nCounter Analysis Software 2.0, and nSolver Advanced Analysis 4.0 (NanoString Technologies, Seattle, WA). The nCounter software detects probes based on a doubling of the counts relative to the median count value of the negative control. Expression data for the 19 genes of interest was normalized using the expression values for the

five internal control genes. Fold changes and p-values were calculated using a Benjamini-Yekutieli (B-Y) correction for multiple comparisons. The B-Y correction assumes that there may be a biological connection between genes and returns moderately conservative estimates of false discovery rate (FDR). The FDR is the proportion of genes with equal or greater evidence for differential expression (i.e., equal or lower raw *p*-value) than are expected to be "false discoveries" due to chance.

Comparisons were made based on farm, disease, and age. First, age matched healthy calves (CRS-/TUS-) were compared between farms A and B. Second, comparisons were made between age matched BRD and healthy calves over the course of disease progression with each BRD group compared against the healthy baseline. The CRS-/TUS+ and CRS+/TUS+ groups were then combined to create a larger TUS+ group and compared to the healthy baseline. Lastly, calves were compared solely by week of age (3, 5, 7, or 9) in groupings of healthy, TUS +, and all calves with the younger age serving as the baseline in each comparison.

## Results

### Calves and clinical outcomes

The calves selected for DGE analysis (n = 29) were either diseased (n = 19) or healthy (n = 10). The diseased calves fit one of three case definitions for BRD at the onset of disease: CRS +/TUS- (n = 7); CRS-/TUS+ (n = 6); or CRS+/TUS+ (n = 6). Three consecutive samples were analyzed for each diseased calf: PRE, ONSET, and POST. The PRE sample was from two weeks prior to the onset of disease when the calf was healthy. The ONSET sample was from when the calf was identified as diseased by one of the three BRD case definitions above. The POST sample was from two weeks after ONSET and was not defined by diagnostic parameters. The numbers of calves with onset of disease in each week were as follows and are shown in Fig 1. Week 5 (n = 10): CRS+/TUS- (n = 4); CRS-/TUS+ (n = 2); and CRS+/TUS+ (n = 4). Week 7 (n = 9): CRS+/TUS- (n = 3); CRS-/TUS+ (n = 4); and CRS+/TUS+ (n = 2). Three or four consecutive samples were also analyzed for each of the healthy calves with the total number of healthy samples in each week as follows: week 3 (n = 9), week 5 (n = 10), week 7 (n = 10), and week 9 (n = 10).

The POST sample time point was not defined by diagnostic parameters, and a variety of disease progression or resolution was observed at this sampling. Additionally, five calves received treatment by farm personnel using the antibiotic florfenicol in combination with the nonsteroidal anti-inflammatory flunixin meglumine (Resflor, Merck Animal Health, Kenilworth, NJ) according to farm protocols between the ONSET and POST sample time points. Of the calves that were CRS+/TUS- at ONSET (n = 7), four resolved to CRS-/TUS-, two remained CRS +/TUS- (one of which was treated with Resflor) but had a score of ≥ 2 in a different one of the three respiratory parameters (nasal, eyes, and cough), and one progressed to CRS- with a TUS score of 2. Of the calves that were CRS-/TUS+ at ONSET (n = 6), four remained CRS-/TUS+, one was treated with Resflor and resolved to CRS-/TUS-, and one progressed to CRS+/TUS-. Of the calves that were CRS+/TUS+ at ONSET (n = 6), two progressed to CRS- with a TUS score of 2 (one of which was treated with Resflor), two progressed to CRS-/TUS+ (one of which was treated with Resflor), and two remained CRS+/TUS+ with one of those having CRS + defined by a different respiratory parameter than at ONSET and the other one receiving treatment with Resflor (S1 Table). Of all the calves that were TUS+ at ONSET (n = 12), eight calves remained TUS+ at POST with at least one lung lobe fully consolidated (two of these having received treatment with Resflor), two calves partially resolved to have lobular consolidation (TUS score of 2; one of these having received treatment with Resflor), and two calves fully resolved their visible lung lesions (TUS-; one of these having received treatment with Resflor).

**Differential gene expression.** RNA integrity number values ranged from 9.1–10.0 (mean = 9.7; StDev = 0.4) for the randomly selected samples, indicating intact RNA and sufficient storage and extraction methods [34]. RNA quality control metrics were assessed for all the samples (n = 96) using nSolver 4.0 (NanoString Technologies, Seattle, WA). None were flagged for imaging, binding density, positive control, or limit of detection; therefore, expression data from all the samples were included in the analyses. The expression of the internal control genes was likewise found to be within the expected range for consistent background expression, and expression data for the 19 genes of interest was normalized using the values for the five internal control genes. Differential expression was analyzed with nSolver's advanced analysis feature using normalized data in a mixture negative binomial model. Low count data was omitted by removing probes that fell below a standard threshold count value of 20, and three genes (*LIF*, *IFNG*, and *TNF*) were omitted from the analysis for this reason.

Age-matched healthy calves on farm A (n = 6) and farm B (n = 4) were compared and showed no DGE between the populations, including in week 5 of life following intranasal vaccination on farm A (B-Y $p > 0.3$). Calves were not differentiated by farm in further analyses. Healthy and BRD calves were matched by age (week 3, 5, 7, or 9) and disease progression (PRE, ONSET, or POST) with healthy calves used as the baseline for DGE analyses. The CRS +/TUS- group compared with the healthy group showed differences in two genes, *ISG15* and *MX1*, in week 7 POST (B-Y $p = 0.04$), with no differential expression observed in the rest of the analysis (B-Y $p > 0.05$). The CRS-/TUS+ and CRS+/TUS+ groups were each likewise compared to the healthy group, and no differential expression was observed in either analysis (B-Y $p > 0.05$). We considered the comparisons involving calves with ultrasonographic evidence of lobar consolidation (TUS+) to be the most meaningful from a clinical perspective since TUS has greater sensitivity and specificity as a diagnostic modality than does clinical scoring for BRD diagnosis [35]. Therefore, CRS-/TUS+ and CRS+/TUS+ classifications were combined to create a larger, more statistically powerful TUS+ group (n = 12) and compared to the healthy baseline. No differential expression was observed in this analysis either (B-Y $p > 0.05$; Table 2).

Calves were compared by week of age as follows: 3 vs. 5, 3 vs. 7, 3 vs. 9, 5 vs. 7, 5 vs. 9, and 7 vs. 9. Age comparisons were made using groupings of healthy (CRS-/TUS-) (Table 3), TUS+ (Table 4), and all calves (Table 5). Gene expression differences were observed for one or more of the genes of interest in all the age comparisons (B-Y $p < 0.05$; Tables 3–5). Consistent trends were observed across age comparisons regarding which genes were up-regulated (*ALOX15*, *HPGD*, *GZMB*, *IL17D*, *LCN2*), down-regulated (*BPI*, *OAS2*, *SELP*, *S100A8*, *ISG15*, *IL1R2*, *DHX58*, *MX1*), or not differentially expressed (*CATHL6*, *PGLYRP1*, *CXCL8*) in the older age samples as compared to younger ages. Relative magnitudes of log fold changes differed for each gene across comparisons; however, there were no instances of a gene being differentially expressed with a positive log fold change in one comparison and a negative log fold change in another. The greatest number of differentially expressed genes were found with the grouping of all calves which also had the most robust sample size. The direction and magnitude of the expression differences across age comparisons for this grouping are presented in Fig 2.

## Discussion

To the authors' knowledge, this is the first study to investigate DGE relative to BRD in pre-weaned Holstein calves. We did not identify differential expression of the genes of interest in age-matched calves relative to lobar lung consolidation as determined via TUS. However, we

**Table 2. Log₂ fold changes in peripheral leukocyte gene expression in Holstein calves with lobar lung consolidation (TUS+; n = 12) as compared to healthy calves (CRS-/TUS-; n = 10).**

| Gene Name | Week | Log₂ fold change | Std error (log₂) | Lower CL (log₂) | Upper CL (log₂) | *P*-value | B-Y *p*-value[a] |
|---|---|---|---|---|---|---|---|
| ALOX15 | 3 Pre | -1.03 | 0.65 | -2.29 | 0.24 | 0.1150 | 1.0000 |
| | 5 Pre | -0.11 | 0.62 | -1.33 | 1.11 | 0.8600 | 1.0000 |
| | 5 Onset | -0.84 | 0.62 | -2.07 | 0.38 | 0.1800 | 1.0000 |
| | 7 Onset | -0.27 | 0.62 | -1.49 | 0.95 | 0.6650 | 1.0000 |
| | 7 Post | -0.38 | 0.62 | -1.60 | 0.84 | 0.5410 | 1.0000 |
| | 9 Post | 0.74 | 0.62 | -0.48 | 1.96 | 0.2380 | 1.0000 |
| BPI | 3 Pre | -0.57 | 0.47 | -1.49 | 0.34 | 0.2240 | 1.0000 |
| | 5 Pre | 0.08 | 0.46 | -0.82 | 0.98 | 0.8640 | 1.0000 |
| | 5 Onset | -0.89 | 0.46 | -1.78 | 0.01 | 0.0566 | 0.5690 |
| | 7 Onset | -0.72 | 0.46 | -1.62 | 0.18 | 0.1200 | 0.9280 |
| | 7 Post | -0.48 | 0.46 | -1.38 | 0.42 | 0.3000 | 1.0000 |
| | 9 Post | -0.02 | 0.46 | -0.91 | 0.88 | 0.9730 | 1.0000 |
| CATHL6 | 3 Pre | 0.36 | 0.22 | -0.07 | 0.78 | 0.1040 | 1.0000 |
| | 5 Pre | 0.16 | 0.22 | -0.26 | 0.59 | 0.4560 | 1.0000 |
| | 5 Onset | -0.08 | 0.22 | -0.51 | 0.36 | 0.7290 | 1.0000 |
| | 7 Onset | 0.43 | 0.21 | 0.02 | 0.85 | 0.0435 | 0.6080 |
| | 7 Post | 0.43 | 0.21 | 0.02 | 0.85 | 0.0435 | 0.7850 |
| | 9 Post | 0.15 | 0.22 | -0.28 | 0.57 | 0.4970 | 1.0000 |
| CXCL8 | 3 Pre | -0.17 | 0.51 | -1.18 | 0.84 | 0.7380 | 1.0000 |
| | 5 Pre | -0.82 | 0.50 | -1.81 | 0.17 | 0.1070 | 1.0000 |
| | 5 Onset | -0.13 | 0.50 | -1.12 | 0.86 | 0.7950 | 1.0000 |
| | 7 Onset | -0.36 | 0.50 | -1.35 | 0.63 | 0.4740 | 1.0000 |
| | 7 Post | -0.07 | 0.50 | -1.05 | 0.92 | 0.8940 | 1.0000 |
| | 9 Post | -0.55 | 0.50 | -1.54 | 0.44 | 0.2780 | 1.0000 |
| DHX58 | 3 Pre | -0.50 | 0.35 | -1.19 | 0.19 | 0.1590 | 1.0000 |
| | 5 Pre | 0.50 | 0.34 | -0.18 | 1.17 | 0.1540 | 1.0000 |
| | 5 Onset | -0.44 | 0.34 | -1.11 | 0.24 | 0.2090 | 1.0000 |
| | 7 Onset | -0.06 | 0.34 | -0.73 | 0.62 | 0.8670 | 1.0000 |
| | 7 Post | -0.14 | 0.34 | -0.82 | 0.53 | 0.6840 | 1.0000 |
| | 9 Post | 0.13 | 0.34 | -0.55 | 0.80 | 0.7150 | 1.0000 |
| GZMB | 3 Pre | -0.91 | 0.37 | -1.63 | -0.18 | 0.0161 | 0.7870 |
| | 5 Pre | -0.34 | 0.36 | -1.05 | 0.37 | 0.3470 | 1.0000 |
| | 5 Onset | -0.75 | 0.36 | -1.46 | -0.05 | 0.0394 | 0.5690 |
| | 7 Onset | -0.74 | 0.36 | -1.45 | -0.04 | 0.0427 | 0.6080 |
| | 7 Post | -0.86 | 0.36 | -1.56 | -0.15 | 0.0194 | 0.5250 |
| | 9 Post | -0.64 | 0.36 | -1.34 | 0.07 | 0.0793 | 1.0000 |
| HPGD | 3 Pre | -0.52 | 0.46 | -1.42 | 0.38 | 0.2570 | 1.0000 |
| | 5 Pre | -0.37 | 0.44 | -1.23 | 0.50 | 0.4090 | 1.0000 |
| | 5 Onset | -0.92 | 0.45 | -1.79 | -0.05 | 0.0412 | 0.5690 |
| | 7 Onset | -0.80 | 0.44 | -1.66 | 0.07 | 0.0737 | 0.6650 |
| | 7 Post | -0.65 | 0.44 | -1.51 | 0.22 | 0.1470 | 1.0000 |
| | 9 Post | 0.56 | 0.44 | -0.31 | 1.42 | 0.2080 | 1.0000 |

*(Continued)*

**Table 2.** (Continued)

| Gene Name | Week | Log$_2$ fold change | Std error (log$_2$) | Lower CL (log$_2$) | Upper CL (log$_2$) | P-value | B-Y p-value[a] |
|---|---|---|---|---|---|---|---|
| IL17D | 3 Pre | 0.12 | 0.29 | -0.45 | 0.68 | 0.6890 | 1.0000 |
| | 5 Pre | 0.41 | 0.26 | -0.10 | 0.93 | 0.1200 | 1.0000 |
| | 5 Onset | 0.18 | 0.27 | -0.35 | 0.70 | 0.5070 | 1.0000 |
| | 7 Onset | -0.03 | 0.26 | -0.55 | 0.48 | 0.8980 | 1.0000 |
| | 7 Post | 0.00 | 0.26 | -0.52 | 0.51 | 0.9950 | 1.0000 |
| | 9 Post | 0.44 | 0.26 | -0.07 | 0.94 | 0.0938 | 1.0000 |
| IL1R2 | 3 Pre | -0.38 | 0.30 | -0.97 | 0.22 | 0.2220 | 1.0000 |
| | 5 Pre | -0.56 | 0.30 | -1.15 | 0.03 | 0.0665 | 0.8990 |
| | 5 Onset | -0.53 | 0.30 | -1.12 | 0.07 | 0.0848 | 0.6550 |
| | 7 Onset | -0.02 | 0.30 | -0.61 | 0.57 | 0.9500 | 1.0000 |
| | 7 Post | -0.98 | 0.30 | -1.57 | -0.38 | 0.0019 | 0.1000 |
| | 9 Post | 0.02 | 0.30 | -0.57 | 0.61 | 0.9510 | 1.0000 |
| ISG15 | 3 Pre | -0.65 | 0.57 | -1.76 | 0.46 | 0.2540 | 1.0000 |
| | 5 Pre | 0.76 | 0.55 | -0.32 | 1.85 | 0.1710 | 1.0000 |
| | 5 Onset | -0.57 | 0.55 | -1.65 | 0.52 | 0.3100 | 1.0000 |
| | 7 Onset | -0.06 | 0.55 | -1.15 | 1.02 | 0.9090 | 1.0000 |
| | 7 Post | -0.53 | 0.55 | -1.61 | 0.56 | 0.3420 | 1.0000 |
| | 9 Post | -0.05 | 0.55 | -1.13 | 1.04 | 0.9320 | 1.0000 |
| LCN2 | 3 Pre | -0.02 | 0.27 | -0.54 | 0.51 | 0.9490 | 1.0000 |
| | 5 Pre | 0.50 | 0.26 | -0.01 | 1.01 | 0.0588 | 0.8990 |
| | 5 Onset | 0.49 | 0.26 | -0.02 | 1.00 | 0.0631 | 0.5690 |
| | 7 Onset | 0.59 | 0.26 | 0.08 | 1.10 | 0.0270 | 0.6080 |
| | 7 Post | 0.03 | 0.26 | -0.49 | 0.54 | 0.9210 | 1.0000 |
| | 9 Post | 0.13 | 0.26 | -0.38 | 0.64 | 0.6170 | 1.0000 |
| MX1 | 3 Pre | -0.88 | 0.40 | -1.66 | -0.10 | 0.0291 | 0.7870 |
| | 5 Pre | 0.51 | 0.39 | -0.25 | 1.27 | 0.1950 | 1.0000 |
| | 5 Onset | -0.63 | 0.39 | -1.39 | 0.13 | 0.1080 | 0.7310 |
| | 7 Onset | 0.15 | 0.39 | -0.61 | 0.91 | 0.7000 | 1.0000 |
| | 7 Post | -0.47 | 0.39 | -1.23 | 0.30 | 0.2330 | 1.0000 |
| | 9 Post | 0.20 | 0.39 | -0.56 | 0.96 | 0.6110 | 1.0000 |
| OAS2 | 3 Pre | -1.00 | 0.52 | -2.03 | 0.02 | 0.0577 | 1.0000 |
| | 5 Pre | 0.05 | 0.51 | -0.96 | 1.05 | 0.9300 | 1.0000 |
| | 5 Onset | -1.00 | 0.51 | -2.00 | 0.00 | 0.0542 | 0.5690 |
| | 7 Onset | -0.13 | 0.51 | -1.13 | 0.87 | 0.7960 | 1.0000 |
| | 7 Post | -0.74 | 0.51 | -1.75 | 0.26 | 0.1510 | 1.0000 |
| | 9 Post | 0.22 | 0.51 | -0.79 | 1.22 | 0.6740 | 1.0000 |
| PGLYRP1 | 3 Pre | 0.04 | 0.25 | -0.45 | 0.52 | 0.8840 | 1.0000 |
| | 5 Pre | 0.06 | 0.24 | -0.42 | 0.54 | 0.8070 | 1.0000 |
| | 5 Onset | 0.01 | 0.24 | -0.47 | 0.49 | 0.9570 | 1.0000 |
| | 7 Onset | 0.26 | 0.24 | -0.21 | 0.73 | 0.2790 | 1.0000 |
| | 7 Post | 0.21 | 0.24 | -0.26 | 0.68 | 0.3890 | 1.0000 |
| | 9 Post | -0.26 | 0.25 | -0.74 | 0.22 | 0.2850 | 1.0000 |

(*Continued*)

**Table 2.** (Continued)

| Gene Name | Week | Log₂ fold change | Std error (log₂) | Lower CL (log₂) | Upper CL (log₂) | P-value | B-Y p-value[a] |
|---|---|---|---|---|---|---|---|
| S100A8 | 3 Pre | -0.01 | 0.31 | -0.61 | 0.60 | 0.9860 | 1.0000 |
| | 5 Pre | 0.58 | 0.30 | -0.01 | 1.16 | 0.0585 | 0.8990 |
| | 5 Onset | 0.68 | 0.30 | 0.09 | 1.27 | 0.0257 | 0.5690 |
| | 7 Onset | 0.61 | 0.30 | 0.02 | 1.20 | 0.0450 | 0.6080 |
| | 7 Post | -0.29 | 0.30 | -0.88 | 0.30 | 0.3440 | 1.0000 |
| | 9 Post | -0.63 | 0.30 | -1.21 | -0.04 | 0.0401 | 1.0000 |
| SELP | 3 Pre | -0.37 | 0.27 | -0.89 | 0.16 | 0.1730 | 1.0000 |
| | 5 Pre | -0.76 | 0.26 | -1.27 | -0.25 | 0.0047 | 0.2520 |
| | 5 Onset | -0.02 | 0.26 | -0.54 | 0.49 | 0.9280 | 1.0000 |
| | 7 Onset | -0.47 | 0.26 | -0.99 | 0.04 | 0.0734 | 0.6650 |
| | 7 Post | -0.18 | 0.26 | -0.69 | 0.34 | 0.5050 | 1.0000 |
| | 9 Post | -0.13 | 0.26 | -0.64 | 0.39 | 0.6310 | 1.0000 |

Comparisons are made between TUS+ calves (any CRS score with a TUS score of ≥ 3 at ONSET) and healthy calves (CRS ≤ 1 in all the respiratory parameters, nasal, eyes, and cough, and TUS ≤ 1). Samples are matched by week of age (3–9) and disease progression time point (PRE, ONSET, or POST). There are twelve TUS+ calves with six having onset of disease in week 5 and six having onset of disease in week 7. The healthy group is comprised of nine individuals in week 3 and those same individuals plus one more for a total of ten in weeks 5, 7, and 9. The healthy calf samples are used as the baseline with the BRD calf samples serving as the comparison. The magnitude and direction of the log fold change can be interpreted as referring to the change in the BRD calves relative to the healthy. Genes IFNG, LIF, and TNF are not assessed due to consistently falling below the recommended expression minimum of 20 copies per sample.

[a]B-Y p-value: Benjamini-Yekutieli method for p-value adjustment.

did observe consistent patterns of differential expression relative to calf age throughout the preweaning period. This suggests that age-related factors and immune system development may be more influential to gene expression in peripheral leukocytes in young calves than inflammatory disease processes.

We considered the TUS+ vs healthy analysis to be the most clinically meaningful of the disease-based analyses. TUS has greater sensitivity and specificity for diagnosis of BRD than does clinical scoring [35]. Although differential expression of *ISG15* and *MX1* was observed in the CRS+/TUS- vs healthy analysis in the week 7 POST comparison, we did not consider this finding to be clinically significant. The POST time point was expected to demonstrate residual effects of the inflammatory event initiated at ONSET, and the interpretation of differential expression at POST in the absence of differential expression at ONSET in terms of inflammatory gene expression differences was unclear. Additionally, we did not observe a consistent progression of clinical respiratory signs from ONSET to POST. Calves that had a nasal, eye, or cough score of 2 at ONSET either had improvement of their clinical signs or had an elevated score in a different one of the respiratory parameters two weeks later at POST. There were no instances of the same respiratory parameter being scored ≥ 2 at both ONSET and POST in the absence of TUS lesions. These findings caused us to suspect that at least some of the clinical respiratory signs we observed were in response to transient environmental conditions or upper respiratory irritation and not indicative of a lower respiratory disease process. Conversely, there was evident continuity between the TUS lesions at ONSET and POST. Although two of the TUS+ calves were resolved at POST, one of these having been treated by farm personnel, the remaining ten remained diseased at the same level, increased in lesion severity, or marginally improved while retaining at least a lobular lesion, and three of these had received treatment (S1 Table).

**Table 3. Log$_2$ fold changes in peripheral leukocyte gene expression in healthy Holstein calves (CRS-/TUS-; n = 10) compared by week of age.**

| Gene Name | Week | Log$_2$ fold change | Std error (log$_2$) | Lower CL (log$_2$) | Upper CL (log$_2$) | P-value | B-Y p-value[a] |
|---|---|---|---|---|---|---|---|
| ALOX15 | 3 vs. 5 | 2.25 | 0.50 | 1.27 | 3.24 | <0.0001 | 0.0012 |
| | 3 vs. 7 | 2.78 | 0.50 | 1.80 | 3.77 | <0.0001 | <0.0001 |
| | 3 vs. 9 | 2.02 | 0.50 | 1.03 | 3.00 | 0.0001 | 0.0044 |
| | 5 vs. 7 | 0.53 | 0.49 | -0.42 | 1.49 | 0.2760 | 1.0000 |
| | 5 vs. 9 | -0.24 | 0.49 | -1.19 | 0.72 | 0.6300 | 1.0000 |
| | 7 vs. 9 | -0.77 | 0.49 | -1.72 | 0.19 | 0.1180 | 1.0000 |
| BPI | 3 vs. 5 | -0.81 | 0.41 | -1.60 | -0.02 | 0.0483 | 0.8210 |
| | 3 vs. 7 | -1.22 | 0.41 | -2.01 | -0.42 | 0.0035 | 0.0267 |
| | 3 vs. 9 | -1.47 | 0.41 | -2.26 | -0.67 | 0.0005 | 0.0054 |
| | 5 vs. 7 | -0.41 | 0.39 | -1.18 | 0.37 | 0.3060 | 1.0000 |
| | 5 vs. 9 | -0.66 | 0.39 | -1.43 | 0.12 | 0.1000 | 0.7750 |
| | 7 vs. 9 | -0.25 | 0.40 | -1.02 | 0.53 | 0.5300 | 1.0000 |
| CATHL6 | 3 vs. 5 | -0.03 | 0.20 | -0.42 | 0.36 | 0.8800 | 1.0000 |
| | 3 vs. 7 | 0.03 | 0.20 | -0.36 | 0.41 | 0.8970 | 1.0000 |
| | 3 vs. 9 | 0.02 | 0.20 | -0.37 | 0.40 | 0.9370 | 1.0000 |
| | 5 vs. 7 | 0.06 | 0.19 | -0.32 | 0.43 | 0.7740 | 1.0000 |
| | 5 vs. 9 | 0.05 | 0.19 | -0.33 | 0.42 | 0.8140 | 1.0000 |
| | 7 vs. 9 | -0.01 | 0.19 | -0.38 | 0.36 | 0.9590 | 1.0000 |
| CXCL8 | 3 vs. 5 | -0.03 | 0.46 | -0.93 | 0.87 | 0.9450 | 1.0000 |
| | 3 vs. 7 | -0.20 | 0.46 | -1.09 | 0.70 | 0.6720 | 1.0000 |
| | 3 vs. 9 | -0.37 | 0.46 | -1.27 | 0.53 | 0.4210 | 1.0000 |
| | 5 vs. 7 | -0.16 | 0.45 | -1.04 | 0.71 | 0.7150 | 1.0000 |
| | 5 vs. 9 | -0.34 | 0.45 | -1.21 | 0.53 | 0.4490 | 1.0000 |
| | 7 vs. 9 | -0.18 | 0.45 | -1.05 | 0.70 | 0.6940 | 1.0000 |
| DHX58 | 3 vs. 5 | -0.17 | 0.33 | -0.81 | 0.48 | 0.6130 | 1.0000 |
| | 3 vs. 7 | -1.00 | 0.33 | -1.64 | -0.35 | 0.0032 | 0.0267 |
| | 3 vs. 9 | -1.02 | 0.33 | -1.67 | -0.37 | 0.0027 | 0.0206 |
| | 5 vs. 7 | -0.83 | 0.32 | -1.46 | -0.20 | 0.0113 | 0.2040 |
| | 5 vs. 9 | -0.85 | 0.32 | -1.48 | -0.22 | 0.0094 | 0.1610 |
| | 7 vs. 9 | -0.02 | 0.32 | -0.65 | 0.61 | 0.9470 | 1.0000 |
| GZMB | 3 vs. 5 | 0.37 | 0.33 | -0.28 | 1.02 | 0.2630 | 1.0000 |
| | 3 vs. 7 | 0.60 | 0.33 | -0.05 | 1.24 | 0.0743 | 0.3350 |
| | 3 vs. 9 | 0.79 | 0.33 | 0.14 | 1.44 | 0.0190 | 0.0935 |
| | 5 vs. 7 | 0.22 | 0.32 | -0.41 | 0.85 | 0.4870 | 1.0000 |
| | 5 vs. 9 | 0.42 | 0.32 | -0.21 | 1.05 | 0.1980 | 1.0000 |
| | 7 vs. 9 | 0.19 | 0.32 | -0.44 | 0.82 | 0.5500 | 1.0000 |
| HPGD | 3 vs. 5 | 0.74 | 0.39 | -0.02 | 1.50 | 0.0607 | 0.8210 |
| | 3 vs. 7 | 1.19 | 0.39 | 0.44 | 1.95 | 0.0027 | 0.0267 |
| | 3 vs. 9 | 0.32 | 0.39 | -0.45 | 1.08 | 0.4190 | 1.0000 |
| | 5 vs. 7 | 0.46 | 0.37 | -0.28 | 1.19 | 0.2250 | 1.0000 |
| | 5 vs. 9 | -0.42 | 0.38 | -1.16 | 0.32 | 0.2660 | 1.0000 |
| | 7 vs. 9 | -0.88 | 0.38 | -1.61 | -0.14 | 0.0216 | 0.4370 |

(*Continued*)

**Table 3.** (Continued)

| Gene Name | Week | Log$_2$ fold change | Std error (log$_2$) | Lower CL (log$_2$) | Upper CL (log$_2$) | P-value | B-Y p-value[a] |
|---|---|---|---|---|---|---|---|
| IL17D | 3 vs. 5 | 0.41 | 0.25 | -0.08 | 0.90 | 0.1070 | 1.0000 |
| | 3 vs. 7 | 0.69 | 0.25 | 0.20 | 1.17 | 0.0068 | 0.0456 |
| | 3 vs. 9 | 0.62 | 0.25 | 0.13 | 1.10 | 0.0146 | 0.0791 |
| | 5 vs. 7 | 0.28 | 0.23 | -0.18 | 0.74 | 0.2380 | 1.0000 |
| | 5 vs. 9 | 0.21 | 0.24 | -0.25 | 0.67 | 0.3740 | 1.0000 |
| | 7 vs. 9 | -0.07 | 0.23 | -0.52 | 0.39 | 0.7680 | 1.0000 |
| IL1R2 | 3 vs. 5 | -0.55 | 0.29 | -1.12 | 0.01 | 0.0586 | 0.8210 |
| | 3 vs. 7 | -0.64 | 0.29 | -1.20 | -0.07 | 0.0302 | 0.1820 |
| | 3 vs. 9 | -0.78 | 0.29 | -1.34 | -0.21 | 0.0083 | 0.0499 |
| | 5 vs. 7 | -0.08 | 0.28 | -0.63 | 0.47 | 0.7690 | 1.0000 |
| | 5 vs. 9 | -0.23 | 0.28 | -0.78 | 0.33 | 0.4220 | 1.0000 |
| | 7 vs. 9 | -0.14 | 0.28 | -0.70 | 0.41 | 0.6100 | 1.0000 |
| ISG15 | 3 vs. 5 | -0.46 | 0.55 | -1.53 | 0.61 | 0.4000 | 1.0000 |
| | 3 vs. 7 | -1.98 | 0.55 | -3.05 | -0.91 | 0.0005 | 0.0126 |
| | 3 vs. 9 | -1.82 | 0.55 | -2.89 | -0.75 | 0.0013 | 0.0114 |
| | 5 vs. 7 | -1.52 | 0.53 | -2.56 | -0.48 | 0.0054 | 0.1450 |
| | 5 vs. 9 | -1.36 | 0.53 | -2.40 | -0.32 | 0.0124 | 0.1610 |
| | 7 vs. 9 | 0.16 | 0.53 | -0.88 | 1.20 | 0.7630 | 1.0000 |
| LCN2 | 3 vs. 5 | 0.34 | 0.26 | -0.17 | 0.85 | 0.1980 | 1.0000 |
| | 3 vs. 7 | 0.56 | 0.26 | 0.05 | 1.08 | 0.0337 | 0.1820 |
| | 3 vs. 9 | 0.97 | 0.26 | 0.46 | 1.48 | 0.0004 | 0.0048 |
| | 5 vs. 7 | 0.23 | 0.25 | -0.27 | 0.72 | 0.3790 | 1.0000 |
| | 5 vs. 9 | 0.63 | 0.25 | 0.13 | 1.13 | 0.0149 | 0.1610 |
| | 7 vs. 9 | 0.41 | 0.25 | -0.09 | 0.90 | 0.1130 | 1.0000 |
| MX1 | 3 vs. 5 | -0.32 | 0.38 | -1.08 | 0.43 | 0.4000 | 1.0000 |
| | 3 vs. 7 | -1.16 | 0.38 | -1.91 | -0.41 | 0.0033 | 0.0267 |
| | 3 vs. 9 | -1.08 | 0.38 | -1.83 | -0.33 | 0.0062 | 0.0416 |
| | 5 vs. 7 | -0.84 | 0.37 | -1.57 | -0.10 | 0.0279 | 0.3780 |
| | 5 vs. 9 | -0.75 | 0.37 | -1.49 | -0.02 | 0.0469 | 0.4220 |
| | 7 vs. 9 | 0.08 | 0.37 | -0.65 | 0.81 | 0.8270 | 1.0000 |
| OAS2 | 3 vs. 5 | -0.16 | 0.51 | -1.15 | 0.83 | 0.7550 | 1.0000 |
| | 3 vs. 7 | -1.77 | 0.51 | -2.77 | -0.79 | 0.0007 | 0.0126 |
| | 3 vs. 9 | -1.93 | 0.51 | -2.92 | -0.94 | 0.0002 | 0.0044 |
| | 5 vs. 7 | -1.62 | 0.49 | -2.58 | -0.65 | 0.0015 | 0.0786 |
| | 5 vs. 9 | -1.78 | 0.49 | -2.74 | -0.81 | 0.0005 | 0.0184 |
| | 7 vs. 9 | -0.16 | 0.49 | -1.12 | 0.81 | 0.7480 | 1.0000 |
| PGLYRP1 | 3 vs. 5 | -0.12 | 0.22 | -0.55 | 0.31 | 0.5740 | 1.0000 |
| | 3 vs. 7 | -0.05 | 0.22 | -0.48 | 0.38 | 0.8200 | 1.0000 |
| | 3 vs. 9 | 0.05 | 0.22 | -0.38 | 0.47 | 0.8220 | 1.0000 |
| | 5 vs. 7 | 0.07 | 0.21 | -0.34 | 0.49 | 0.7320 | 1.0000 |
| | 5 vs. 9 | 0.17 | 0.21 | -0.24 | 0.59 | 0.4200 | 1.0000 |
| | 7 vs. 9 | 0.10 | 0.21 | -0.32 | 0.51 | 0.6420 | 1.0000 |

(*Continued*)

**Table 3.** (Continued)

| Gene Name | Week | Log$_2$ fold change | Std error (log$_2$) | Lower CL (log$_2$) | Upper CL (log$_2$) | *P*-value | B-Y *p*-value[a] |
|---|---|---|---|---|---|---|---|
| S100A8 | 3 vs. 5 | -0.32 | 0.27 | -0.85 | 0.20 | 0.2320 | 1.0000 |
| | 3 vs. 7 | -0.54 | 0.27 | -1.06 | -0.01 | 0.0478 | 0.2350 |
| | 3 vs. 9 | 0.06 | 0.27 | -0.47 | 0.59 | 0.8230 | 1.0000 |
| | 5 vs. 7 | -0.22 | 0.26 | -0.73 | 0.30 | 0.4110 | 1.0000 |
| | 5 vs. 9 | 0.38 | 0.26 | -0.13 | 0.90 | 0.1460 | 0.9860 |
| | 7 vs. 9 | 0.60 | 0.26 | 0.09 | 1.11 | 0.0242 | 0.4370 |
| SELP | 3 vs. 5 | -0.11 | 0.25 | -0.60 | 0.37 | 0.6480 | 1.0000 |
| | 3 vs. 7 | -0.17 | 0.25 | -0.65 | 0.32 | 0.5020 | 1.0000 |
| | 3 vs. 9 | -0.97 | 0.25 | -1.45 | -0.48 | 0.0002 | 0.0044 |
| | 5 vs. 7 | -0.05 | 0.24 | -0.53 | 0.42 | 0.8250 | 1.0000 |
| | 5 vs. 9 | -0.85 | 0.24 | -1.33 | -0.38 | 0.0007 | 0.0184 |
| | 7 vs. 9 | -0.80 | 0.24 | -1.27 | -0.32 | 0.0014 | 0.0754 |

Comparisons are made between ages of healthy calves (CRS $\leq$ 1 in all the respiratory parameters, nasal, eyes, and cough, and TUS $\leq$ 1) using serial samples. There are nine total individuals in week 3 and those same individuals plus one more for a total of ten in weeks 5, 7, and 9. The younger age samples are used as the baseline with the older age samples serving as the comparison. The magnitude and direction of the log fold change can be interpreted as referring to the change in the older sample relative to the younger. Genes IFNG, LIF, and TNF are not assessed due to consistently falling below the recommended expression minimum of 20 copies per sample.

[a]B-Y p-value: Benjamini-Yekutieli method for p-value adjustment.

The genes of interest in this study were based on previous research in postweaned or adult cattle. Multiple studies in feedlot cattle with BRD have identified common differentially expressed genes associated with disease presence, severity, and outcomes [14, 15, 19, 21]. Likewise, studies aimed at identifying biomarkers of diseases affecting adult dairy cows such as mastitis, metritis, and Johne's disease have identified consistent gene expression changes associated with subclinical or clinical disease that may bridge disease states [28–30]. The differential expression patterns observed in these studies involving older animals were not observed in our cohort of pre-weaned calves, suggesting that there are additional factors affecting leukocyte behavior in young calves.

Early life shifts in calf leukocyte populations are a likely contributing factor to the age-related gene expression changes we observed. In a study of Norwegian Red calves, substantial changes in leukocyte cell type absolute numbers, lymphocyte subpopulation proportions, and neutrophil functions occurred in the first 5–8 weeks of life with ongoing changes up to 6 months of age [36]. More recently, 30-day old Holstein calves were shown to have distinctly different lymphocyte subpopulation proportions as compared to adults including fewer B cells and comparatively more γδ T cells [37]. Furthermore, monocyte subpopulations have been found to be phenotypically and functionally distinct between 10-day old and 18–24 month old Holstein-Friesian cattle both at baseline and following an in vitro *Neospora caninum* challenge [38]. Given these early life changes, it is unsurprising that leukocyte gene expression profiles of young calves would differ from those expected in more mature animals in response to an inflammatory disease process such as BRD.

These differences in leukocyte profiles between calves and mature cattle are also the likely reason that three of our genes of interest (*LIF*, *IFNG*, and *TNF*) had consistently low copy numbers in our samples and were unable to be reliably included in DGE analysis. Although these genes are well documented inflammatory mediators in older cattle [14, 17, 29, 30], they are evidently not prolifically expressed by peripheral leukocytes of young calves.

**Table 4. Log₂ fold changes in peripheral leukocyte gene expression in Holstein calves with lobar lung consolidation (TUS+; n = 12) compared by week of age.**

| Gene Name | Week | Log₂ fold change | Std error (log₂) | Lower CL (log₂) | Upper CL (log₂) | *P*-value | B-Y *p*-value[a] |
|---|---|---|---|---|---|---|---|
| ALOX15 | 3 vs. 5 | 2.85 | 0.61 | 1.64 | 4.05 | <0.0001 | 0.0007 |
| | 3 vs. 7 | 3.49 | 0.61 | 2.28 | 4.69 | <0.0001 | <0.0001 |
| | 3 vs. 9 | 3.78 | 0.71 | 2.40 | 5.17 | <0.0001 | <0.0001 |
| | 5 vs. 7 | 0.64 | 0.50 | -0.33 | 1.61 | 0.2010 | 1.0000 |
| | 5 vs. 9 | 0.94 | 0.61 | -0.25 | 2.12 | 0.1270 | 0.8630 |
| | 7 vs. 9 | 0.30 | 0.61 | -0.89 | 1.49 | 0.6240 | 1.0000 |
| BPI | 3 vs. 5 | -0.56 | 0.45 | -1.45 | 0.32 | 0.2140 | 1.0000 |
| | 3 vs. 7 | -1.24 | 0.45 | -2.12 | -0.36 | 0.0073 | 0.0980 |
| | 3 vs. 9 | -0.91 | 0.52 | -1.93 | 0.11 | 0.0839 | 0.5680 |
| | 5 vs. 7 | -0.68 | 0.37 | -1.40 | 0.05 | 0.0699 | 0.5400 |
| | 5 vs. 9 | -0.35 | 0.45 | -1.23 | 0.54 | 0.4450 | 1.0000 |
| | 7 vs. 9 | 0.33 | 0.45 | -0.56 | 1.21 | 0.4670 | 1.0000 |
| CATHL6 | 3 vs. 5 | -0.34 | 0.21 | -0.74 | 0.07 | 0.1040 | 0.9390 |
| | 3 vs. 7 | 0.10 | 0.20 | -0.30 | 0.50 | 0.6220 | 1.0000 |
| | 3 vs. 9 | -0.20 | 0.24 | -0.66 | 0.27 | 0.4160 | 1.0000 |
| | 5 vs. 7 | 0.44 | 0.17 | 0.11 | 0.77 | 0.0109 | 0.1180 |
| | 5 vs. 9 | 0.15 | 0.21 | -0.27 | 0.56 | 0.4920 | 1.0000 |
| | 7 vs. 9 | -0.30 | 0.21 | -0.70 | 0.11 | 0.1560 | 1.0000 |
| CXCL8 | 3 vs. 5 | -0.30 | 0.49 | -1.26 | 0.67 | 0.5510 | 1.0000 |
| | 3 vs. 7 | -0.23 | 0.49 | -1.19 | 0.73 | 0.6420 | 1.0000 |
| | 3 vs. 9 | -0.75 | 0.57 | -1.86 | 0.37 | 0.1910 | 1.0000 |
| | 5 vs. 7 | 0.07 | 0.40 | -0.72 | 0.85 | 0.8720 | 1.0000 |
| | 5 vs. 9 | -0.45 | 0.49 | -1.42 | 0.51 | 0.3590 | 1.0000 |
| | 7 vs. 9 | -0.52 | 0.49 | -1.48 | 0.45 | 0.2950 | 1.0000 |
| DHX58 | 3 vs. 5 | 0.44 | 0.34 | -0.24 | 1.11 | 0.2070 | 1.0000 |
| | 3 vs. 7 | -0.60 | 0.34 | -1.27 | 0.07 | 0.0848 | 0.4590 |
| | 3 vs. 9 | -0.39 | 0.40 | -1.17 | 0.38 | 0.3220 | 1.0000 |
| | 5 vs. 7 | -1.03 | 0.28 | -1.58 | -0.49 | 0.0004 | 0.0070 |
| | 5 vs. 9 | -0.83 | 0.34 | -1.50 | -0.16 | 0.0175 | 0.2400 |
| | 7 vs. 9 | 0.20 | 0.34 | -0.47 | 0.88 | 0.5540 | 1.0000 |
| GZMB | 3 vs. 5 | 0.75 | 0.35 | 0.06 | 1.44 | 0.0369 | 0.3990 |
| | 3 vs. 7 | 0.71 | 0.35 | 0.01 | 1.39 | 0.0485 | 0.2920 |
| | 3 vs. 9 | 1.06 | 0.41 | 0.26 | 1.85 | 0.0109 | 0.1170 |
| | 5 vs. 7 | -0.04 | 0.29 | -0.60 | 0.52 | 0.8840 | 1.0000 |
| | 5 vs. 9 | 0.31 | 0.35 | -0.38 | 1.00 | 0.3790 | 1.0000 |
| | 7 vs. 9 | 0.35 | 0.35 | -0.34 | 1.04 | 0.3180 | 1.0000 |
| HPGD | 3 vs. 5 | 0.64 | 0.44 | -0.22 | 1.50 | 0.1460 | 0.9890 |
| | 3 vs. 7 | 1.00 | 0.44 | 0.14 | 1.85 | 0.0249 | 0.2070 |
| | 3 vs. 9 | 1.40 | 0.50 | 0.42 | 2.38 | 0.0065 | 0.0878 |
| | 5 vs. 7 | 0.36 | 0.35 | -0.34 | 1.05 | 0.3170 | 1.0000 |
| | 5 vs. 9 | 0.76 | 0.43 | -0.09 | 1.60 | 0.0821 | 0.7400 |
| | 7 vs. 9 | 0.40 | 0.43 | -0.44 | 1.24 | 0.3520 | 1.0000 |

(*Continued*)

**Table 4.** (Continued)

| Gene Name | Week | Log$_2$ fold change | Std error (log$_2$) | Lower CL (log$_2$) | Upper CL (log$_2$) | P-value | B-Y p-value[a] |
|---|---|---|---|---|---|---|---|
| IL17D | 3 vs. 5 | 0.59 | 0.27 | 0.07 | 1.11 | 0.0282 | 0.3810 |
| | 3 vs. 7 | 0.55 | 0.27 | 0.03 | 1.07 | 0.0405 | 0.2740 |
| | 3 vs. 9 | 0.94 | 0.30 | 0.35 | 1.53 | 0.0023 | 0.0420 |
| | 5 vs. 7 | -0.04 | 0.21 | -0.45 | 0.37 | 0.8490 | 1.0000 |
| | 5 vs. 9 | 0.35 | 0.25 | -0.14 | 0.84 | 0.1690 | 1.0000 |
| | 7 vs. 9 | 0.39 | 0.25 | -0.10 | 0.88 | 0.1260 | 1.0000 |
| IL1R2 | 3 vs. 5 | -0.72 | 0.30 | -1.31 | -0.13 | 0.0192 | 0.3460 |
| | 3 vs. 7 | -0.68 | 0.30 | -1.27 | -0.09 | 0.0267 | 0.2070 |
| | 3 vs. 9 | -0.39 | 0.35 | -1.07 | 0.30 | 0.2710 | 1.0000 |
| | 5 vs. 7 | 0.04 | 0.25 | -0.45 | 0.53 | 0.8720 | 1.0000 |
| | 5 vs. 9 | 0.33 | 0.30 | -0.26 | 0.93 | 0.2720 | 1.0000 |
| | 7 vs. 9 | 0.29 | 0.30 | -0.30 | 0.89 | 0.3340 | 1.0000 |
| ISG15 | 3 vs. 5 | 0.43 | 0.55 | -0.64 | 1.51 | 0.4310 | 1.0000 |
| | 3 vs. 7 | -1.61 | 0.55 | -2.68 | -0.54 | 0.0042 | 0.0764 |
| | 3 vs. 9 | -1.22 | 0.63 | -2.46 | 0.02 | 0.0573 | 0.4420 |
| | 5 vs. 7 | -2.04 | 0.45 | -2.92 | -1.17 | <0.0001 | 0.0009 |
| | 5 vs. 9 | -1.65 | 0.55 | -2.73 | -0.58 | 0.0034 | 0.1270 |
| | 7 vs. 9 | 0.39 | 0.55 | -0.68 | 1.46 | 0.4780 | 1.0000 |
| LCN2 | 3 vs. 5 | 0.85 | 0.26 | 0.35 | 1.36 | 0.0014 | 0.0372 |
| | 3 vs. 7 | 0.91 | 0.26 | 0.41 | 1.42 | 0.0006 | 0.0169 |
| | 3 vs. 9 | 1.12 | 0.30 | 0.54 | 1.70 | 0.0003 | 0.0082 |
| | 5 vs. 7 | 0.06 | 0.21 | -0.35 | 0.47 | 0.7660 | 1.0000 |
| | 5 vs. 9 | 0.27 | 0.26 | -0.24 | 0.77 | 0.3010 | 1.0000 |
| | 7 vs. 9 | 0.20 | 0.26 | -0.30 | 0.71 | 0.4280 | 1.0000 |
| MX1 | 3 vs. 5 | 0.61 | 0.39 | -0.16 | 1.37 | 0.1260 | 0.9760 |
| | 3 vs. 7 | -0.40 | 0.39 | -1.17 | 0.36 | 0.3060 | 1.0000 |
| | 3 vs. 9 | 0.00 | 0.45 | -0.89 | 0.89 | 0.9960 | 1.0000 |
| | 5 vs. 7 | -1.01 | 0.32 | -1.64 | -0.38 | 0.0022 | 0.0300 |
| | 5 vs. 9 | -0.60 | 0.39 | -1.37 | 0.17 | 0.1280 | 0.8630 |
| | 7 vs. 9 | 0.41 | 0.39 | -0.36 | 1.17 | 0.3030 | 1.0000 |
| OAS2 | 3 vs. 5 | 0.46 | 0.51 | -0.53 | 1.45 | 0.3630 | 1.0000 |
| | 3 vs. 7 | -1.18 | 0.51 | -2.17 | -0.19 | 0.0220 | 0.2070 |
| | 3 vs. 9 | -0.71 | 0.58 | -1.86 | 0.43 | 0.2240 | 1.0000 |
| | 5 vs. 7 | -1.64 | 0.41 | -2.45 | -0.83 | 0.0001 | 0.0039 |
| | 5 vs. 9 | -1.18 | 0.51 | -2.16 | -0.19 | 0.0222 | 0.2400 |
| | 7 vs. 9 | 0.46 | 0.51 | -0.53 | 1.45 | 0.3620 | 1.0000 |
| PGLYRP | 3 vs. 5 | -0.12 | 0.24 | -0.58 | 0.34 | 0.6030 | 1.0000 |
| | 3 vs. 7 | 0.15 | 0.23 | -0.31 | 0.61 | 0.5230 | 1.0000 |
| | 3 vs. 9 | -0.25 | 0.27 | -0.79 | 0.29 | 0.3620 | 1.0000 |
| | 5 vs. 7 | 0.27 | 0.19 | -0.10 | 0.65 | 0.1580 | 1.0000 |
| | 5 vs. 9 | -0.13 | 0.24 | -0.59 | 0.34 | 0.5930 | 1.0000 |
| | 7 vs. 9 | -0.40 | 0.24 | -0.86 | 0.06 | 0.0942 | 1.0000 |

(*Continued*)

**Table 4.** (Continued)

| Gene Name | Week | Log$_2$ fold change | Std error (log$_2$) | Lower CL (log$_2$) | Upper CL (log$_2$) | *P*-value | B-Y *p*-value[a] |
|---|---|---|---|---|---|---|---|
| S100A8 | 3 vs. 5 | 0.31 | 0.30 | -0.28 | 0.90 | 0.3020 | 1.0000 |
| | 3 vs. 7 | -0.30 | 0.30 | -0.89 | 0.29 | 0.3190 | 1.0000 |
| | 3 vs. 9 | -0.56 | 0.35 | -1.24 | 0.12 | 0.1100 | 0.6640 |
| | 5 vs. 7 | -0.61 | 0.25 | -1.10 | -0.13 | 0.0143 | 0.1290 |
| | 5 vs. 9 | -0.87 | 0.30 | -1.46 | -0.28 | 0.0047 | 0.1270 |
| | 7 vs. 9 | -0.26 | 0.30 | -0.85 | 0.33 | 0.3920 | 1.0000 |
| SELP | 3 vs. 5 | -0.09 | 0.26 | -0.61 | 0.42 | 0.7230 | 1.0000 |
| | 3 vs. 7 | -0.12 | 0.26 | -0.63 | 0.40 | 0.6530 | 1.0000 |
| | 3 vs. 9 | -0.73 | 0.30 | -1.32 | -0.13 | 0.0187 | 0.1680 |
| | 5 vs. 7 | -0.03 | 0.21 | -0.44 | 0.39 | 0.9070 | 1.0000 |
| | 5 vs. 9 | -0.63 | 0.26 | -1.15 | -0.12 | 0.0180 | 0.2400 |
| | 7 vs. 9 | -0.61 | 0.26 | -1.12 | -0.09 | 0.0229 | 1.0000 |

Comparisons are made between ages of TUS+ calves (any CRS score with a TUS score of ≥ 3 at ONSET; n = 12) using serial samples. The younger age samples are used as the baseline with the older age samples serving as the comparison. The magnitude and direction of the log fold change can be interpreted as referring to the change in the older sample relative to the younger. Genes IFNG, LIF, and TNF are not assessed due to consistently falling below the recommended expression minimum of 20 copies per sample.

[a]B-Y p-value: Benjamini-Yekutieli method for p-value adjustment.

Gene expression differences between neonates and immunologically mature animals are supported by studies in human medical fields. Olin et al. [25] analyzed serial samples of peripheral blood leukocytes from babies throughout the first 3 months of life and found substantial age-related phenotypic and transcriptomic changes whereas adult parameters remained relatively stable over a similar period. Additionally, pediatric immune system development occurred on a shared trajectory despite considerable differences in pathology, maturity at birth, environment, and other variables. Similarly, Wynn et al. [24] demonstrated age-related transcriptomic changes in pediatric patients with the degree of difference increasing in proportion to age difference and with the neonatal group being the most distinctive.

Developmental differences in leukocyte gene expression, surface protein profiles, and functionality make pre-weaned calves immunologically distinct from adult cattle. Indeed, the rate and magnitude of change in the early weeks of life arguably prevent pre-weaned calves from being considered comparable as a group with respect to these parameters. This developmental homeorhesis is the likely reason for our failure to demonstrate differential expression of genes of interest relative to the presence of BRD as was expected based on prior findings in older cattle.

This developmental progression may also account for the expression trend we observed with the number of differentially expressed genes and the magnitude of the difference being greatest with younger samples and with increasing difference in age (Fig 2). This expression trend was clearly observed with respect to *ALOX15* which was strongly differentially expressed in comparisons of week 3 to older weeks in our three age-related analyses, and to a lesser extent *HPGD* in the comparison of all calves by week of age (Tables 3–5, Fig 2). Additionally, a marked increase in log fold-change was observed from the 3 vs. 5 to the 3 vs. 7 comparisons across the analyses (Tables 3–5). This trend continued from the 3 vs. 7 to the 3 vs. 9 comparisons in the analysis of TUS+ calves (Table 4).

*ALOX15* encodes an enzyme involved in production of specialized pro-resolving mediators (SPMs). This class of bioactive lipids is instrumental to the resolution phase of the

**Table 5. Log₂ fold changes in peripheral leukocyte gene expression in Holstein calves with and without clinical (CRS) or ultrasonographic (TUS) evidence of respiratory disease (n = 29) compared by week of age.**

| Gene Name | Week | Log$_2$ fold change | Std error (log$_2$) | Lower CL (log$_2$) | Upper CL (log$_2$) | P-value | B-Y p-value[a] |
|---|---|---|---|---|---|---|---|
| ALOX15 | 3 vs. 5 | 2.37 | 0.33 | 1.72 | 3.02 | <0.0001 | <0.0001 |
| | 3 vs. 7 | 3.43 | 0.33 | 2.78 | 4.08 | <0.0001 | <0.0001 |
| | 3 vs. 9 | 2.69 | 0.37 | 1.97 | 3.41 | <0.0001 | <0.0001 |
| | 5 vs. 7 | 1.06 | 0.29 | 0.49 | 1.64 | 0.0005 | 0.0071 |
| | 5 vs. 9 | 0.32 | 0.33 | -0.32 | 0.97 | 0.3300 | 1.0000 |
| | 7 vs. 9 | -0.74 | 0.33 | -1.39 | -0.10 | 0.0269 | 0.4850 |
| BPI | 3 vs. 5 | -0.83 | 0.26 | -1.34 | -0.31 | 0.0021 | 0.0383 |
| | 3 vs. 7 | -1.42 | 0.26 | -1.93 | -0.90 | <0.0001 | <0.0001 |
| | 3 vs. 9 | -1.41 | 0.29 | -1.97 | -0.85 | <0.0001 | 0.0001 |
| | 5 vs. 7 | -0.59 | 0.23 | -1.05 | -0.13 | 0.0130 | 0.0880 |
| | 5 vs. 9 | -0.58 | 0.26 | -1.10 | -0.07 | 0.0280 | 0.1690 |
| | 7 vs. 9 | 0.01 | 0.26 | -0.51 | 0.52 | 0.9830 | 1.0000 |
| CATHL6 | 3 vs. 5 | -0.19 | 0.13 | -0.45 | 0.06 | 0.1440 | 0.9750 |
| | 3 vs. 7 | 0.02 | 0.13 | -0.24 | 0.27 | 0.8900 | 1.0000 |
| | 3 vs. 9 | -0.17 | 0.14 | -0.45 | 0.11 | 0.2400 | 0.9290 |
| | 5 vs. 7 | 0.21 | 0.12 | -0.02 | 0.44 | 0.0739 | 0.4440 |
| | 5 vs. 9 | 0.02 | 0.13 | -0.24 | 0.28 | 0.8650 | 1.0000 |
| | 7 vs. 9 | -0.19 | 0.13 | -0.44 | 0.07 | 0.1550 | 1.0000 |
| CXCL8 | 3 vs. 5 | -0.13 | 0.30 | -0.71 | 0.46 | 0.6760 | 1.0000 |
| | 3 vs. 7 | -0.16 | 0.30 | -0.74 | 0.43 | 0.6060 | 1.0000 |
| | 3 vs. 9 | -0.29 | 0.33 | -0.94 | 0.35 | 0.3760 | 1.0000 |
| | 5 vs. 7 | -0.03 | 0.27 | -0.55 | 0.49 | 0.9120 | 1.0000 |
| | 5 vs. 9 | -0.17 | 0.30 | -0.76 | 0.42 | 0.5790 | 1.0000 |
| | 7 vs. 9 | -0.14 | 0.30 | -0.73 | 0.45 | 0.6470 | 1.0000 |
| DHX58 | 3 vs. 5 | 0.17 | 0.22 | -0.25 | 0.60 | 0.4300 | 1.0000 |
| | 3 vs. 7 | -0.54 | 0.22 | -0.96 | -0.11 | 0.0152 | 0.0750 |
| | 3 vs. 9 | -0.58 | 0.24 | -1.05 | -0.12 | 0.0163 | 0.0802 |
| | 5 vs. 7 | -0.71 | 0.19 | -1.09 | -0.33 | 0.0004 | 0.0071 |
| | 5 vs. 9 | -0.76 | 0.22 | -1.18 | -0.33 | 0.0008 | 0.0103 |
| | 7 vs. 9 | -0.05 | 0.22 | -0.47 | 0.38 | 0.8290 | 1.0000 |
| GZMB | 3 vs. 5 | 0.61 | 0.22 | 0.18 | 1.04 | 0.0071 | 0.0742 |
| | 3 vs. 7 | 0.90 | 0.22 | 0.47 | 1.33 | 0.0001 | 0.0014 |
| | 3 vs. 9 | 1.12 | 0.24 | 0.64 | 1.59 | <0.0001 | 0.0002 |
| | 5 vs. 7 | 0.29 | 0.20 | -0.10 | 0.67 | 0.1440 | 0.7100 |
| | 5 vs. 9 | 0.51 | 0.22 | 0.08 | 0.94 | 0.0236 | 0.1590 |
| | 7 vs. 9 | 0.22 | 0.22 | -0.21 | 0.65 | 0.3240 | 1.0000 |
| HPGD | 3 vs. 5 | 0.83 | 0.26 | 0.33 | 1.33 | 0.0017 | 0.0383 |
| | 3 vs. 7 | 1.55 | 0.26 | 1.06 | 2.05 | <0.0001 | <0.0001 |
| | 3 vs. 9 | 0.83 | 0.28 | 0.28 | 1.38 | 0.0040 | 0.0243 |
| | 5 vs. 7 | 0.73 | 0.23 | 0.29 | 1.17 | 0.0016 | 0.0176 |
| | 5 vs. 9 | 0.00 | 0.25 | -0.49 | 0.50 | 0.9930 | 1.0000 |
| | 7 vs. 9 | -0.73 | 0.25 | -1.22 | -0.23 | 0.0050 | 0.1350 |

*(Continued)*

**Table 5.** (Continued)

| Gene Name | Week | Log$_2$ fold change | Std error (log$_2$) | Lower CL (log$_2$) | Upper CL (log$_2$) | P-value | B-Y p-value[a] |
|---|---|---|---|---|---|---|---|
| IL17D | 3 vs. 5 | 0.35 | 0.16 | 0.03 | 0.67 | 0.0330 | 0.2550 |
| | 3 vs. 7 | 0.58 | 0.16 | 0.26 | 0.89 | 0.0005 | 0.0048 |
| | 3 vs. 9 | 0.71 | 0.18 | 0.37 | 1.06 | <0.0001 | 0.0011 |
| | 5 vs. 7 | 0.23 | 0.14 | -0.05 | 0.50 | 0.1100 | 0.5950 |
| | 5 vs. 9 | 0.36 | 0.16 | 0.06 | 0.67 | 0.0221 | 0.1590 |
| | 7 vs. 9 | 0.14 | 0.15 | -0.17 | 0.44 | 0.3770 | 1.0000 |
| IL1R2 | 3 vs. 5 | -0.52 | 0.19 | -0.88 | -0.15 | 0.0067 | 0.0742 |
| | 3 vs. 7 | -0.68 | 0.19 | -1.05 | -0.32 | 0.0004 | 0.0046 |
| | 3 vs. 9 | -0.72 | 0.21 | -1.12 | -0.32 | 0.0007 | 0.0052 |
| | 5 vs. 7 | -0.17 | 0.17 | -0.49 | 0.16 | 0.3220 | 1.0000 |
| | 5 vs. 9 | -0.21 | 0.19 | -0.57 | 0.16 | 0.2750 | 1.0000 |
| | 7 vs. 9 | -0.04 | 0.19 | -0.41 | 0.33 | 0.8320 | 1.0000 |
| ISG1 | 3 vs. 5 | 0.17 | 0.36 | -0.54 | 0.88 | 0.6360 | 1.0000 |
| | 3 vs. 7 | -0.99 | 0.36 | -1.70 | -0.28 | 0.0077 | 0.0417 |
| | 3 vs. 9 | -1.14 | 0.40 | -1.92 | -0.36 | 0.0053 | 0.0284 |
| | 5 vs. 7 | -1.16 | 0.32 | -1.79 | -0.53 | 0.0005 | 0.0071 |
| | 5 vs. 9 | -1.31 | 0.36 | -2.02 | -0.60 | 0.0005 | 0.0100 |
| | 7 vs. 9 | -0.15 | 0.36 | -0.86 | 0.56 | 0.6760 | 1.0000 |
| LCN2 | 3 vs. 5 | 0.46 | 0.17 | 0.13 | 0.79 | 0.0082 | 0.0742 |
| | 3 vs. 7 | 0.60 | 0.17 | 0.27 | 0.93 | 0.0006 | 0.0048 |
| | 3 vs. 9 | 0.88 | 0.19 | 0.51 | 1.24 | <0.0001 | 0.0002 |
| | 5 vs. 7 | 0.14 | 0.15 | -0.15 | 0.44 | 0.3450 | 1.0000 |
| | 5 vs. 9 | 0.42 | 0.17 | 0.09 | 0.75 | 0.0147 | 0.1320 |
| | 7 vs. 9 | 0.28 | 0.17 | -0.05 | 0.61 | 0.1040 | 1.0000 |
| MX1 | 3 vs. 5 | 0.14 | 0.26 | -0.36 | 0.64 | 0.5970 | 1.0000 |
| | 3 vs. 7 | -0.46 | 0.26 | -0.96 | 0.04 | 0.0765 | 0.3450 |
| | 3 vs. 9 | -0.56 | 0.28 | -1.11 | -0.01 | 0.0494 | 0.2230 |
| | 5 vs. 7 | -0.59 | 0.23 | -1.04 | -0.15 | 0.0105 | 0.0815 |
| | 5 vs. 9 | -0.69 | 0.26 | -1.19 | -0.19 | 0.0078 | 0.0843 |
| | 7 vs. 9 | -0.10 | 0.26 | -0.60 | 0.40 | 0.6920 | 1.0000 |
| OAS2 | 3 vs. 5 | 0.11 | 0.33 | -0.54 | 0.77 | 0.7360 | 1.0000 |
| | 3 vs. 7 | -1.11 | 0.33 | -1.77 | -0.46 | 0.0012 | 0.0074 |
| | 3 vs. 9 | -1.08 | 0.37 | -1.80 | -0.36 | 0.0041 | 0.0243 |
| | 5 vs. 7 | -1.23 | 0.30 | -1.81 | -0.64 | <0.0001 | 0.0043 |
| | 5 vs. 9 | -1.20 | 0.33 | -1.85 | -0.54 | 0.0006 | 0.0100 |
| | 7 vs. 9 | 0.03 | 0.33 | -0.62 | 0.69 | 0.9250 | 1.0000 |
| PGLYRP1 | 3 vs. 5 | -0.18 | 0.14 | -0.46 | 0.10 | 0.2030 | 1.0000 |
| | 3 vs. 7 | -0.08 | 0.14 | -0.36 | 0.20 | 0.5680 | 1.0000 |
| | 3 vs. 9 | -0.14 | 0.16 | -0.45 | 0.16 | 0.3560 | 1.0000 |
| | 5 vs. 7 | 0.10 | 0.13 | -0.15 | 0.35 | 0.4270 | 1.0000 |
| | 5 vs. 9 | 0.04 | 0.14 | -0.24 | 0.32 | 0.7930 | 1.0000 |
| | 7 vs. 9 | -0.06 | 0.14 | -0.34 | 0.21 | 0.6560 | 1.0000 |

(*Continued*)

**Table 5.** (Continued)

| Gene Name | Week | Log₂ fold change | Std error (log₂) | Lower CL (log₂) | Upper CL (log₂) | *P*-value | B-Y *p*-value[a] |
|---|---|---|---|---|---|---|---|
| S100A8 | 3 vs. 5 | -0.11 | 0.18 | -0.46 | 0.25 | 0.5560 | 1.0000 |
| | 3 vs. 7 | -0.61 | 0.18 | -0.96 | -0.25 | 0.0012 | 0.0074 |
| | 3 vs. 9 | -0.32 | 0.20 | -0.71 | 0.07 | 0.1140 | 0.4740 |
| | 5 vs. 7 | -0.50 | 0.16 | -0.82 | -0.18 | 0.0026 | 0.0232 |
| | 5 vs. 9 | -0.21 | 0.18 | -0.57 | 0.14 | 0.2470 | 1.0000 |
| | 7 vs. 9 | 0.29 | 0.18 | -0.07 | 0.64 | 0.1140 | 1.0000 |
| SELP | 3 vs. 5 | -0.10 | 0.16 | -0.42 | 0.22 | 0.5410 | 1.0000 |
| | 3 vs. 7 | -0.10 | 0.16 | -0.42 | 0.22 | 0.5320 | 1.0000 |
| | 3 vs. 9 | -0.70 | 0.18 | -1.05 | -0.34 | 0.0002 | 0.0019 |
| | 5 vs. 7 | 0.00 | 0.15 | -0.29 | 0.28 | 0.9870 | 1.0000 |
| | 5 vs. 9 | -0.60 | 0.16 | -0.92 | -0.27 | 0.0005 | 0.0100 |
| | 7 vs. 9 | -0.59 | 0.16 | -0.91 | -0.27 | 0.0005 | 0.0264 |

Comparisons are made between ages of all calves irrespective of disease state using serial samples. The number of individuals comprising each week are as follows: 19 in week 3, 29 in weeks 5 and 7, and 19 in week 9. The younger age samples are used as the baseline with the older age samples serving as the comparison. The magnitude and direction of the log fold change can be interpreted as referring to the change in the older sample relative to the younger. Genes IFNG, LIF, and TNF are not assessed due to consistently falling below the recommended expression minimum of 20 copies per sample.

[a]B-Y p-value: Benjamini-Yekutieli method for p-value adjustment.

inflammatory cascade and return of the affected tissue to homeostasis through the activity of M2 macrophages [39–41]. In beef calves SPM expression, including *ALOX15* and *HPGD*, has been shown to increase consistently throughout the pre-weaning period in calves undergoing a variety of management and vaccination programs [42]. This aligns with our findings suggesting that changes in SPM expression are an integral part of calf immune development during early life. Consequently, as biomarkers of inflammatory disease these genes are unlikely to be as reliable in pre-weaned calves as in immunologically mature cattle.

Overall, this study of pre-weaned Holstein heifer calves with BRD did not demonstrate notable differential expression of 19 genes that are candidate biomarkers of inflammatory disease in postweaned and adult cattle, and that in some cases are indicative of resistance to or tolerance of disease. Transcriptomic profiling provides a plausible alternative for identifying innate immune system genes and pathways that are differentially expressed with respect to young calf BRD. Given that biomarkers may not be stable throughout the pre-weaning period, narrow age ranges may be necessary to observe consistent results. Furthermore, knowledge of gene expression profiles in healthy pre-weaned calves is lacking and will be necessary to evaluate disease-related changes. Preweaned calves are immunologically distinct from older cattle, and baseline values established in adults must be applied with caution to younger age groups.

## Conclusions

Nineteen genes that are candidate biomarkers of inflammatory disease in postweaned and adult cattle were not differentially expressed in pre-weaned Holstein heifer calves with BRD. However, differential expression was observed relative to calf age between three and nine weeks of life. Factors related to age and immune system development overshadow disease impacts to influence gene expression patterns in young calves, and immune development progresses upon a common trajectory during the preweaning period regardless of respiratory disease.

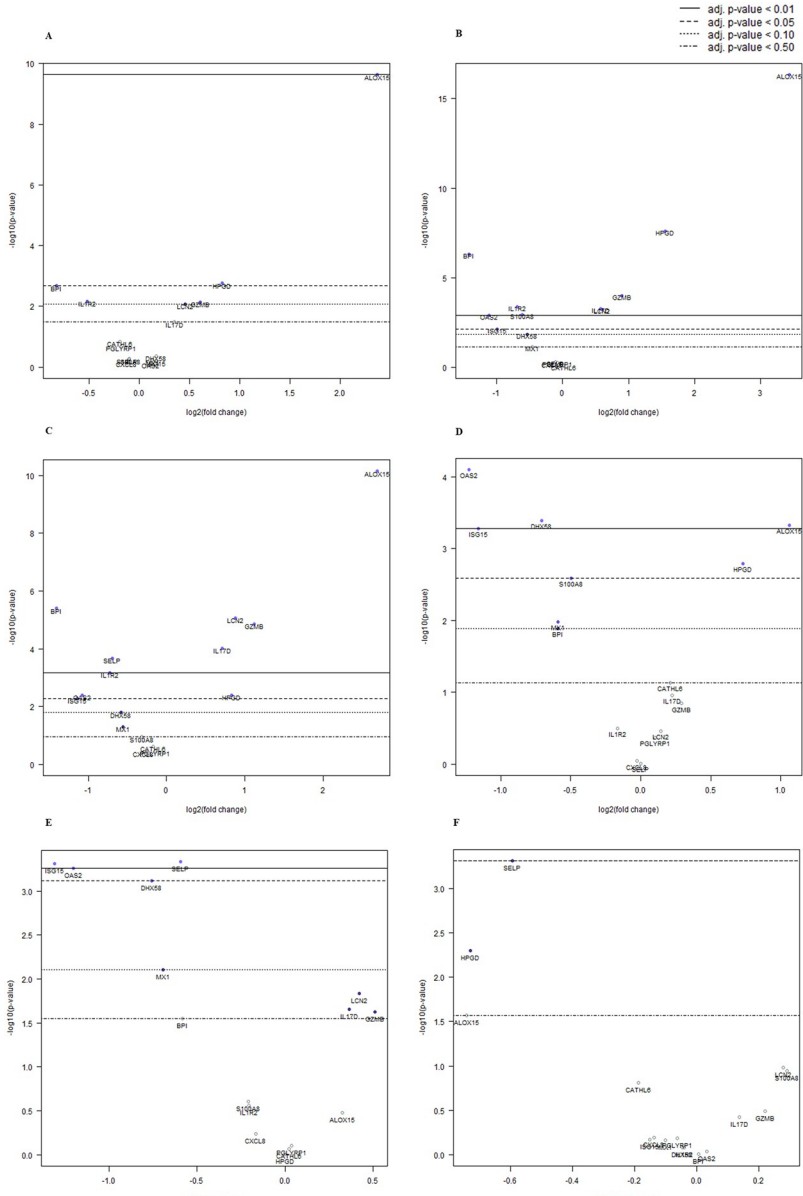

**Fig 2. Differential gene expression magnitude and direction for calves with and without clinical (CRS) or ultrasonographic (TUS) evidence of respiratory disease (n = 29) compared by week of age.** Volcano plots showing differential gene expression between calf ages. (A) week 5 vs. baseline week 3. (B) week 7 vs. baseline week 3. (C) week 9 vs. baseline week 3. (D) week 7 vs. baseline week 5. (E) week 9 vs. baseline week 5. (F) week 9 vs. baseline week 7. In each plot the x-axis is the log$_2$ fold change of the comparison (older) group relative to the baseline (younger) group. Zero indicates no difference between the groups with the positive or negative values indicating increased or decreased expression of the gene in the comparison group relative to the baseline group. The y-axis represents statistical significance with the horizontal lines representing various B-Y p-value cut-offs. Adj. p-value: Benjamini-Yekutieli (B-Y) method for p-value adjustment.

## Supporting information

**S1 Table. Individual CRS and TUS scores for each calf by sampling week.**
(XLSX)

**S2 Table. Raw counts for endogenous, housekeeping, negative, and positive probes.**
(XLSX)

**S3 Table. Probe sequences.**
(XLSX)

## Acknowledgments

The authors thank the participating dairies and associated personnel for their invaluable assistance with this project.

## Author Contributions

**Conceptualization:** Lily A. Elder, Holly R. Hinnant, Craig S. McConnel.

**Data curation:** Lily A. Elder, Craig S. McConnel.

**Formal analysis:** Lily A. Elder, Craig S. McConnel.

**Funding acquisition:** Craig S. McConnel.

**Investigation:** Lily A. Elder, Holly R. Hinnant, Chris M. Mandella, Rachel A. Claus-Walker, Lindsay M. Parrish, Giovana S. Slanzon, Craig S. McConnel.

**Methodology:** Lily A. Elder, Lindsay M. Parrish, Craig S. McConnel.

**Project administration:** Craig S. McConnel.

**Resources:** Craig S. McConnel.

**Supervision:** Craig S. McConnel.

**Writing – original draft:** Lily A. Elder, Craig S. McConnel.

**Writing – review & editing:** Holly R. Hinnant, Chris M. Mandella, Rachel A. Claus-Walker, Lindsay M. Parrish, Giovana S. Slanzon.

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
