## [Decision Letter · Decision Letter 0]

20 Apr 2023

PONE-D-23-08494Differential gene expression in peripheral leukocytes of pre-weaned Holstein heifer calves with respiratory diseasePLOS ONE

Dear Dr. McConnel,

Thank you for submitting your manuscript to PLOS ONE. After careful consideration, we feel that it has merit but does not fully meet PLOS ONE’s publication criteria as it currently stands. Therefore, we invite you to submit a revised version of the manuscript that addresses the points raised during the review process.

We look forward to receiving your revised manuscript.

Kind regards,

Angel Abuelo, DVM, MRes, MSc, PhD, DABVP (Dairy), DECBHM

Academic Editor

PLOS ONE

Journal Requirements:

Additional Editor Comments:

Both reviewers have raised important concerns that preclude the manuscript's acceptance as it stands. Please revise accordingly ensuring that you address all reviewers' points.

Reviewers' comments:

Reviewer's Responses to Questions

**Comments to the Author**

1. Is the manuscript technically sound, and do the data support the conclusions?

Reviewer #1: Partly

Reviewer #2: Yes

2. Has the statistical analysis been performed appropriately and rigorously? 

Reviewer #1: Yes

Reviewer #2: Yes

3. Have the authors made all data underlying the findings in their manuscript fully available?

Reviewer #1: Yes

Reviewer #2: Yes

4. Is the manuscript presented in an intelligible fashion and written in standard English?

Reviewer #1: Yes

Reviewer #2: Yes

5. Review Comments to the Author

Reviewer #1: This manuscript analyzed a panel of 19 genes as potential biomarkers for prediction of BRD. Authors compared pre, onset and post gene expression from whole blood samples in preweaned heifer calves across time. While they did not identify any differences between BRD cases (regardless of diagnoses), they noted changes in gene expression over time.

The paper is clearly written and the study is easy to follow. Although it is largely negative data, the results are relevant for future studies aimed at identifying biomarkers of disease in very young calves. A few comments:

1) Calves in the study received an intranasal vaccine. The vaccine was different and received at different time points. However, there is no discussion or consideration of the effects of the vaccine on gene expression. Because the authors determined early on that GE did not differ, this possibility was disregarded. I think some comparisons should be made to determine if there are effects of the vaccine and/or the different vaccines given at different ages.

2) The authors indicate a major heat event during the course of the study, but do not comment on the timing of this event. A clearer indication of when the heat event (and thus possible heat stress) occurred is important. Were both farms equally effected? Approximately how old were the animals and were major changes observed? this could be buried in the 'age' comparisons but likely has a major impact on interpretation.

3) Figure 1 is not mentioned in the text at all. Figure 2 is not discussed at all in the results section (only discussion). The quality of Figure 2 is very poor and there is no way to evaluate the usefulness of this figure or what it's telling us.

4) Authors mention treatment events in some BRD calves but not others. Were any effects of treatment considered (grouping as +/- treatment, rather than BRD as defined by study personnel)?

Reviewer #2: Thank you for the opportunity to review the manuscript titled “Differential gene expression in peripheral leukocytes of pre-weaned Holstein heifer calves with respiratory disease”. This manuscript performed a controlled time-course analysis of healthy and diseased heifer Holstein calves with NanoString nCounter methodology, further categorizing concurrent respiratory disease via semi-objective scoring assessment and transthoracic ultrasonography. I commend the authors for their novel approach and thoroughness regarding sampling, clinical assessment, and workflow. In its current state, the presented manuscript is well written and of high quality, and I recommend it for publication pending minor revisions and clarification. Below are comments which may provide further insights and improvements to the manuscript.

• Line 219-220: while the Benjamini-Yekutieli correction method for controlling type I error is appropriate, it may be argued that family-wise error correction methodology and false-discovery rates are too conservative due to the limited “independent” hypotheses being tested in nCounter datasets when compared to microarray or next-generation sequencing (DOI: 10.4137/CIN.S16343). This may be an analytical component as to why so few differences were found with these comparisons; however, this approach also provides a strength in demonstrating the differential expression of ALOX15.

• Lines 217-219 & 270-277: the authors should expand on normalization and analysis techniques. What type of nCounter analyzer was used (SPRINT, Pro?)? What was the maximum FOV of detection? How was codeset normalization performed (geometric mean of positive controls? Mean/median/max of negative control counts?)? How was differential gene expression evaluated (Fisher’s exact, Wilcoxon-Rank, etc.)? The aforementioned citation [29] does not provide detail regarding these elements of the project.

• Lines 228-231 & 278-280: I commend the authors for exploring possible effects between farms and rationalizing the lack of need for its use as a random effect in their testing parameters.

• Lines 382-387: I commend the authors on their discussion regarding the potential transient disease that may have occurred within this study; this speaks to the multifaceted nature, and frustration regarding research, of bovine respiratory disease. However, as stated by the authors, these cattle are of different age, breed, and systems, and it may be that the genes selected better represent a more infectious course of respiratory disease, especially of cattle placed into riskier environments (e.g., salebarns, feedlots, etc.). Could it be hypothesized that these cattle, even across two separate farms, were experiencing non-viral (ISG15, MX1, OAS2) or non-infectious (CATHL6, S100A8) course of disease, and that young Holsteins require their own, separate candidate biomarker identification research? Possibly, these distinct gene expression patterns (as detailed by the provided citations) and their associated mechanisms in association with BRD are specific to post-weaned beef systems.

6. PLOS authors have the option to publish the peer review history of their article (what does this mean?). If published, this will include your full peer review and any attached files.

Reviewer #1: No

Reviewer #2: No

---

## [Author Response · Author response to Decision Letter 0]

28 Apr 2023

Response to reviewers

Thank you for your insightful questions and comments. Your commentary has led to changes that we believe make the manuscript a stronger addition to the literature. Please find responses to specific questions below.

Reviewer #1: This manuscript analyzed a panel of 19 genes as potential biomarkers for prediction of BRD. Authors compared pre, onset and post gene expression from whole blood samples in preweaned heifer calves across time. While they did not identify any differences between BRD cases (regardless of diagnoses), they noted changes in gene expression over time.

The paper is clearly written and the study is easy to follow. Although it is largely negative data, the results are relevant for future studies aimed at identifying biomarkers of disease in very young calves. 

A few comments:

1) Calves in the study received an intranasal vaccine. The vaccine was different and received at different time points. However, there is no discussion or consideration of the effects of the vaccine on gene expression. Because the authors determined early on that GE did not differ, this possibility was disregarded. I think some comparisons should be made to determine if there are effects of the vaccine and/or the different vaccines given at different ages. 

AU: Thank you for bringing up the potential for the difference in vaccination protocols between farms impacting calf gene expression. Further analysis aimed at determining if there are gene expression effects of intranasal vaccination, differing by type of vaccine and timing of administration, was not performed for the following reasons. 

Age matched healthy calves were compared between the farms, and no differential gene expression (DGE) was observed (lines 284-286). These comparisons consisted of samples from weeks 3, 5, 7, and 9 of calf life. Intranasal vaccination was done at four weeks of age on farm A, and at two weeks of age on farm B. Although our sampling did not span the timing around vaccination on farm B, no effects were observed in this analysis following vaccination on farm A. Additional text highlighting lack of effect of vaccination has been added on lines 285-286. 

Unfortunately, due to the difference in timing of vaccine administration between farms, a comparison of calves at the post-vaccination time point, week 3 on farm B and week 5 on farm A, would require comparison of calves of different ages. We have found age, irrespective of farm, to be a major source of differential gene expression (lines 313-370), and, therefore, this analysis would be unrewarding for the purpose of identifying effects of vaccination. Additionally, all calves on both farms were vaccinated per farm protocols leaving no unvaccinated individuals to serve as a control group for an analysis of gene expression in age matched vaccinated vs. unvaccinated calves over time. Furthermore, both of the vaccine brands used by the farms are intranasal, modified live virus vaccines. Intranasal vaccination is known to produce a primarily local mucosal response. We are not aware of studies assessing systemic gene expression relative to mucosal vaccination. We recognize that there may have been a transient systemic inflammatory response to intranasal vaccination in the study calves, and this response may have been identifiable in a study utilizing transcriptomic techniques, but given our targeted gene set and conservative statistical approach effects of vaccination were not observed. 

2) The authors indicate a major heat event during the course of the study, but do not comment on the timing of this event. A clearer indication of when the heat event (and thus possible heat stress) occurred is important. Were both farms equally effected? Approximately how old were the animals and were major changes observed? this could be buried in the 'age' comparisons but likely has a major impact on interpretation.

AU: We appreciate the request for clarification regarding the heat event that occurred during our sampling period resulting in hyperthermic but otherwise healthy calves. Farms A and B are located approximately 3 miles from each other and so were equally affected by local weather. There was pervasive heat in the region, outside of the thermoneutral zone, from late June through mid-July primarily corresponding with weeks 6-9 of life of the study calves. Major changes in the calves other than rectal temperature were not observed based on the parameters that we assessed. Additional text has been added to clarify the heat exposure across farms in lines 139-140. 

3) Figure 1 is not mentioned in the text at all. Figure 2 is not discussed at all in the results section (only discussion). The quality of Figure 2 is very poor and there is no way to evaluate the usefulness of this figure or what it's telling us. 

AU: Please see line 166 for the reference to Figure 1 and thank you for bringing your concerns to our attention regarding the image quality of Figure 2. There was debate among the authors about whether or not to include this figure in the manuscript. The data it represents is presented on lines 313-325, shown in Table 5, and discussed on lines 439-447. The decision was ultimately made to include the figure in order to provide a visual demonstration of differential gene expression relative to calf age over time and highlight the notable changes in expression magnitude of several of our genes of interest during the pre-weaning period. We also considered the possibility of only including a single volcano plot for the sake of increasing the size of the figure details; however, we concluded that a lone plot would not be informative and the series of six volcano plots is necessary to tell the whole story. 

We are aware that the PDF version of Figure 2 is of poor quality. However, if it is downloaded as a TIF file it has much better resolution, and zooming in on the individual volcano plots allows thorough evaluation of the figure. We would prefer to leave Figure 2 in the paper given that the downloaded version is appropriately clear, digital publication will allow readers to zoom in on the plots, and there is utility in a visual demonstration when discussing the magnitude of DGE relative to calf age changing over time. 

4) Authors mention treatment events in some BRD calves but not others. Were any effects of treatment considered (grouping as +/- treatment, rather than BRD as defined by study personnel)?

AU: Your point about consideration of treatment in conjunction with BRD in the DGE analysis is well taken. This study was designed with the intention to eliminate treatment as a variable prior to the onset of BRD in order to assess DGE in the absence of antimicrobial or anti-inflammatory drug effects. Although some of the study calves were treated between the ONSET and POST sampling time points, our methods allowed us to identify and sample calves at ONSET prior to any treatment.

An extensive explanation of the variability of clinical signs and ultrasonographic lesions as well as treatments occurring at the POST time point can be found on lines 252-270. Of the 19 BRD calves, the five that received treatment by farm personnel between ONSET and POST were assigned case definitions across all three BRD categories (CRS+/TUS-; CRS-/TUS+; CRS+/TUS+), and each individual had a different combination of CRS and TUS scores at POST. Consequently, an analysis of the POST time point comparing treated vs. untreated calves while taking into account the effects of disease would be unrewarding due to the exceedingly low sample size of treated calves that would be included in each comparison. 

We agree that systemic treatment with a combination of antimicrobial and anti-inflammatory drugs may have an effect at some level on gene expression, and this was a point of discussion among the authors. As with intranasal vaccination, this response may have been identifiable in a study utilizing transcriptomic techniques. However, our finding of no DGE relative to disease state minimizes our concern for this effect substantially impacting the results of this study. Additionally, the PRE and ONSET sampling time points, which we consider to be the most meaningful aspects of the BRD related analysis, were not influenced by any treatment effects. 

Reviewer #2: Thank you for the opportunity to review the manuscript titled “Differential gene expression in peripheral leukocytes of pre-weaned Holstein heifer calves with respiratory disease”. This manuscript performed a controlled time-course analysis of healthy and diseased heifer Holstein calves with NanoString nCounter methodology, further categorizing concurrent respiratory disease via semi-objective scoring assessment and transthoracic ultrasonography. I commend the authors for their novel approach and thoroughness regarding sampling, clinical assessment, and workflow. In its current state, the presented manuscript is well written and of high quality, and I recommend it for publication pending minor revisions and clarification. Below are comments which may provide further insights and improvements to the manuscript.

• Line 219-220: while the Benjamini-Yekutieli correction method for controlling type I error is appropriate, it may be argued that family-wise error correction methodology and false-discovery rates are too conservative due to the limited “independent” hypotheses being tested in nCounter datasets when compared to microarray or next-generation sequencing (DOI:10.4137/CIN.S16343). This may be an analytical component as to why so few differences were found with these comparisons; however, this approach also provides a strength in demonstrating the differential expression of ALOX15.

AU: We appreciate your thoughtful assessment of our statistical methods. We agree that although a less conservative approach may have allowed for additional differences in gene expression to be identified, the conservative approach afforded by the use of the B-Y p-value adjustment likely led to more trustworthy, defensible results such as those found with ALOX15 and reduced the potential noise produced by field conditions. 

• Lines 217-219 & 270-277: the authors should expand on normalization and analysis techniques. What type of nCounter analyzer was used (SPRINT, Pro?)? What was the maximum FOV of detection? How was codeset normalization performed (geometric mean of positive controls? Mean/median/max of negative control counts?)? How was differential gene expression evaluated (Fisher’s exact, Wilcoxon-Rank, etc.)? The aforementioned citation [29] does not provide detail regarding these elements of the project. 

AU: Thank you for your comments regarding the need for additional details about the gene expression normalization and analysis techniques. Test has been included within lines 213-218 and 280-281 to explain that the nCounter run was performed at NanoString using a MAX analyzer, with 555 maximum possible fields of view. There were two consecutive normalizations: 1) A Positive Control Normalization took into account the linearity of the positive controls. The geometric mean of the positive controls was used to compute the normalization factor. 2) CodeSet Content (HouseKeeping Normalization) used the geometric mean of our designated housekeepers to compute a normalization factor. Differential gene expression was evaluated within nSolver advanced analysis using a mixture negative binomial model and a Benjamini-Yekutieli p-value adjustment.

• Lines 228-231 & 278-280: I commend the authors for exploring possible effects between farms and rationalizing the lack of need for its use as a random effect in their testing parameters.

AU: Thank you. We appreciate the commendation. 

• Lines 382-387: I commend the authors on their discussion regarding the potential transient disease that may have occurred within this study; this speaks to the multifaceted nature, and frustration regarding research, of bovine respiratory disease. However, as stated by the authors, these cattle are of different age, breed, and systems, and it may be that the genes selected better represent a more infectious course of respiratory disease, especially of cattle placed into riskier environments (e.g., salebarns, feedlots, etc.). Could it be hypothesized that these cattle, even across two separate farms, were experiencing non-viral (ISG15, MX1, OAS2) or non-infectious (CATHL6, S100A8) course of disease, and that young Holsteins require their own, separate candidate biomarker identification research? Possibly, these distinct gene expression patterns (as detailed by the provided citations) and their associated mechanisms in association with BRD are specific to post-weaned beef systems. 

AU: Thank you for these comments. It seems that we are in agreement about the substantial differences between pre-weaned dairy calves and post-weaned beef cattle regarding candidate BRD biomarkers and the necessity for targeted research in young dairy calves. A combination of age, breed, and rearing system appears to have an impact on peripheral leukocyte gene expression in cattle, and caution is necessary in inferring findings from one population to another in the face of differences in these areas. As you speculated, the disease etiology, pathogens, and pathophysiology experienced by the animals may also impact gene expression. We look forward to continuing this area of study through a follow up transcriptomic study as well as specific BRD pathogen identification on these farms to provide additional detail about the disease processes calves are experiencing in these systems.

---

## [Decision Letter · Decision Letter 1]

4 May 2023

Differential gene expression in peripheral leukocytes of pre-weaned Holstein heifer calves with respiratory disease

PONE-D-23-08494R1

Dear Dr. McConnel,

We’re pleased to inform you that your manuscript has been judged scientifically suitable for publication and will be formally accepted for publication once it meets all outstanding technical requirements.

Kind regards,

Angel Abuelo, DVM, MRes, MSc, PhD, DABVP (Dairy), DECBHM

Academic Editor

PLOS ONE

Additional Editor Comments (optional):

Reviewers' comments:

Reviewer's Responses to Questions

**Comments to the Author**

1. If the authors have adequately addressed your comments raised in a previous round of review and you feel that this manuscript is now acceptable for publication, you may indicate that here to bypass the “Comments to the Author” section, enter your conflict of interest statement in the “Confidential to Editor” section, and submit your "Accept" recommendation.

Reviewer #1: All comments have been addressed

Reviewer #2: All comments have been addressed

2. Is the manuscript technically sound, and do the data support the conclusions?

Reviewer #1: Yes

Reviewer #2: Yes

3. Has the statistical analysis been performed appropriately and rigorously? 

Reviewer #1: Yes

Reviewer #2: Yes

4. Have the authors made all data underlying the findings in their manuscript fully available?

Reviewer #1: Yes

Reviewer #2: Yes

5. Is the manuscript presented in an intelligible fashion and written in standard English?

Reviewer #1: Yes

Reviewer #2: Yes

6. Review Comments to the Author

Reviewer #1: (No Response)

Reviewer #2: The authors have addressed all relevant points in their letter to the reviewers and additions within the revised manuscript. At this time, I recommend the revised manuscript be published in its current state.

7. PLOS authors have the option to publish the peer review history of their article (what does this mean?). If published, this will include your full peer review and any attached files.

Reviewer #1: No

Reviewer #2: No

---

## [Editor Report · Acceptance letter]

8 May 2023

PONE-D-23-08494R1 

Differential gene expression in peripheral leukocytes of pre-weaned Holstein heifer calves with respiratory disease 

Dear Dr. McConnel:

I'm pleased to inform you that your manuscript has been deemed suitable for publication in PLOS ONE. Congratulations! Your manuscript is now with our production department. 

Kind regards, 

on behalf of

Dr. Angel Abuelo 

Academic Editor

PLOS ONE